# Observation of structural switch in nascent SAM-VI riboswitch during transcription at single-nucleotide and single-molecule resolution

Yanyan Xue [1,6], Jun Li [2,6], Dian Chen [1], Xizhu Zhao [3], Liang Hong [2,4,5] & Yu Liu [1,5]

Growing RNAs fold differently as they are transcribed, which modulates their finally adopted structures. Riboswitches regulate gene expression by structural change, which are sensitive to co-transcriptionally structural biology. Here we develop a strategy to track the structural change of RNAs during transcription at single-nucleotide and single-molecule resolution and use it to monitor individual transcripts of the SAM-VI riboswitch (riboSAM) as transcription proceeds, observing co-existence of five states in riboSAM. We report a bifurcated helix in one newly identified state from NMR and single-molecule FRET (smFRET) results, and its presence directs the translation inhibition in our cellular translation experiments. A model is proposed to illustrate the distinct switch patterns and gene-regulatory outcome of riboSAM when SAM is present or absent. Our strategy enables the precise mapping of RNAs' conformational landscape during transcription, and may combine with detection methods other than smFRET for structural studies of RNAs in general.

RNAs' structures during transcription influence their final structures, cellular functions, assembly and molecular interactions[1–9]. However, our understanding of RNAs' structural change at single-molecule resolution during transcription is very limited mainly because only a few methods can achieve such determination. Frieda et al. creatively applied the optical-trapping assay to examine the co-transcriptional folding of adenine riboswitch at single-molecule level[10–12]. Watters et al. combined the chemical probing method with high-throughput sequencing smartly to detect the folding of fluoride and guanine riboswitches at ensemble level[1,13,14]. Delicate transcription complexes assembled with 5′-labeled RNA were used to monitor the co-transcriptional folding of TPP and preQ$_1$ riboswitches by smFRET (single-molecule Förster resonance energy transfer)[15,16]. Helicases have been used to dissociate an RNA-DNA hybrid to mimic the transcription

elongation of the twister ribozyme and ZTP riboswitch by Hua et al.[17,18]. Currently, these reported methods have not yet realized to characterize structures of RNAs when they are synthesized by RNA polymerase at both single-nucleotide and single-molecule levels.

In this work, we modified PLOR (Position-specific Labeling of RNA) method, so that it can synthesize labeled RNAs for post-transcriptional assays, as previously designed[19,20], but importantly, also can combine with smFRET to detect folding of a single RNA molecule at single-nucleotide resolution as transcription proceeds. More specifically, on one hand, we applied PLOR to introduce fluorophores and isotopes to chosen positions of riboSAM to facilitate post-transcriptional studies, including steady-state fluorescence titration, stopped-flow kinetics, smFRET, and NMR. This is achieved by immobilizing biotin-DNA on streptavidin-coated beads, pausing

[1]State Key Laboratory of Microbial Metabolism, School of Life Sciences and Biotechnology, Shanghai Jiao Tong University, Shanghai 200240, China. [2]School of Physics and Astronomy, Shanghai Jiao Tong University, Shanghai 200240, China. [3]Zhiyuan College, Shanghai Jiao Tong University, Shanghai 200240, China. [4]Institute of Natural Sciences, Shanghai Jiao Tong University, Shanghai 200240, China. [5]Shanghai Artificial Intelligence Laboratory, Shanghai 200232, China. [6]These authors contributed equally: Yanyan Xue, Jun Li. ✉e-mail: hongl3liang@sjtu.edu.cn; liuyu_sjtu@sjtu.edu.cn

transcription by omitting certain type(s) of NTPs needed for transcription beyond a specific position, and then restarting transcription by adding a new NTP mix with the required NTP(s). Such pause-restart pattern enables labeling of RNAs at specific positions. And the high-affinity binding between biotin and streptavidin is favorable for minimizing the dissociation of transcription elongation complexes (ECs), which enables to label RNAs with high efficiency. On the other hand, in the modified version of PLOR used here, weaker binding between DNA and streptavidin-coated beads is achieved by replacing biotin-DNA with desthiobiotin-DNA[21,22]. This ensures sufficient ECs dissociate from the beads for the single-molecule FRET determination of nascent RNAs. Moreover, in our strategy, the nascent RNAs can elongate at single-nucleotide pace under physiological conditions (at 2 mM $Mg^{2+}$ and 37 °C).

Riboswitches can regulate gene expression by switching structures after binding to their specific ligands[23–26]. For example, S-adenosyl-L-methionine (SAM) is a well-known methyl donor for many methyl transfer reactions in cells[26–28], and it can also bind specifically to SAM riboswitches and regulate downstream gene expression[29–37]. The SAM-VI riboswitch was identified in 2018, and it was reported to regulate the translation of SAM synthetase after binding with SAM[29]. Folding of riboswitches during transcription can affect their ligand binding and accessibility of ribosomes or polymerases, which is crucial for riboswitches to make gene regulatory decisions[1,5–7]. Currently, structural analyses on the SAM-VI riboswitch (riboSAM) have been focused on post-transcriptional structures, with no study on its structural change during transcription. Here, we applied an adapted strategy to explore the distinct switch patterns of nascent riboSAM with and without SAM in transcription. Based on our results, the structural switch of riboSAM takes place within a narrow transcriptional window

whether SAM exists or not. However, by adding SAM, the switch occurs two nucleotides in advance with higher proportions of SAM-bound structures. Furthermore, the removal of SAM may fold SAM-bound structures that have been formed in the early transcription back to SAM-unbound structures. Interestingly, at least five structures appear during the transcription of riboSAM, three of which have never been reported. It is speculated that a unique bifurcated helix exists in an unreported structure, *transit* by NMR and smFRET analyses. And the *transit* inhibits translation after binding with SAM in our cellular translation assays. A model of conformational switch for nascent riboSAM with and without SAM at single-nucleotide and single-molecule resolution is proposed. Besides coupling with FRET, our strategy can be combined with other methods for the analysis of RNAs as transcription proceeds when appropriate labels are incorporated.

## Results

### The design of determining structural switch of RNAs in transcription

The diagram of our strategy to study the structural change of RNAs in transcription at single-nucleotide and single-molecule levels is presented in Fig. 1. Transcription (following gray arrows) is carried out in the solid phase by immobilizing desthiobiotin-DNA on streptavidin-coated agarose beads (Fig. 1a and Supplementary Fig. 1). Replacing streptavidin-biotin with a weaker coupling, streptavidin-desthiobiotin, increases the dissociation of ECs from the beads (Supplementary Fig. 2). $Mg^{2+}$ is known to impact RNA folding, and the physiological concentrations of $Mg^{2+}$ are 0.5–4 mM[38–40]. Fortunately, in the case of riboSAM, ECs significantly dissociated from the beads at 2 mM $Mg^{2+}$ and 37 °C, amenable for structural determination of nascent RNAs under physiological conditions (Supplementary Fig. 2).

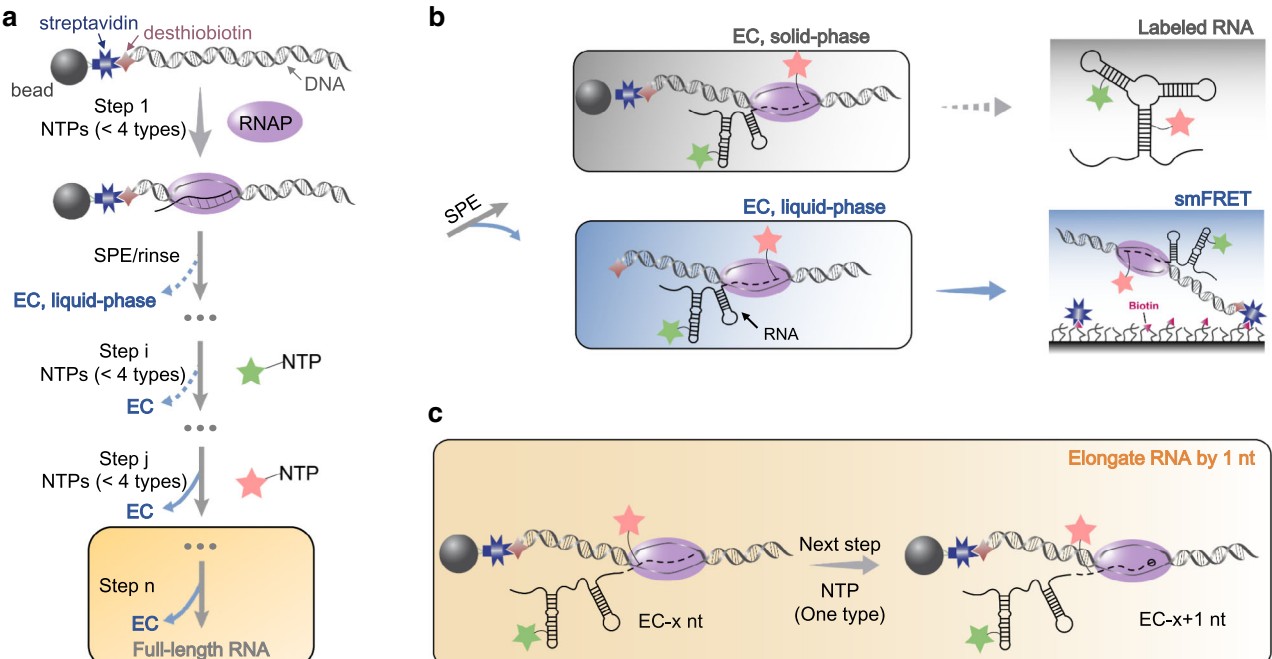

**Fig. 1 | Schematic illustration of co-transcriptional study of ECs labeled with a FRET pair at specific positions. a** The hybrid phase transcription is divided into n steps by a pause-restart mode (following gray arrows), with labels (green and pink stars) incorporated into specific positions of RNA at steps i and j. The pause and restart at each step are precisely controlled by additions of limited NTPs to the solid phase ECs. The 5′-desthiobiotin-labeled DNA templates are coupled to streptavidin-coated agarose beads. The solid phase EC contains DNA templates (gray ribbon), RNAP (purple sphere), nascent transcript (black line) and streptavidin-bead (gray sphere). During transcription, partial ECs are dissociated from beads, generating liquid-phase ECs and being removed from the transcriptional system by SPE (blue arrows). At the end of transcription, the liquid phase contains the full-length labeled RNA used for post-transcriptional assays. **b** After SPE, the solid-phase ECs continue to transcribe and synthesize full-length RNA with labels at step n (gray highlighted). The dissociated ECs collected after step j can be used for co-transcriptional study by smFRET if the labels introduced to ECs are a FRET pair, such as Cy3/Cy5. **c** Single-nucleotide elongation of nascent transcripts in ECs can be achieved by adding one type of NTP in a step.

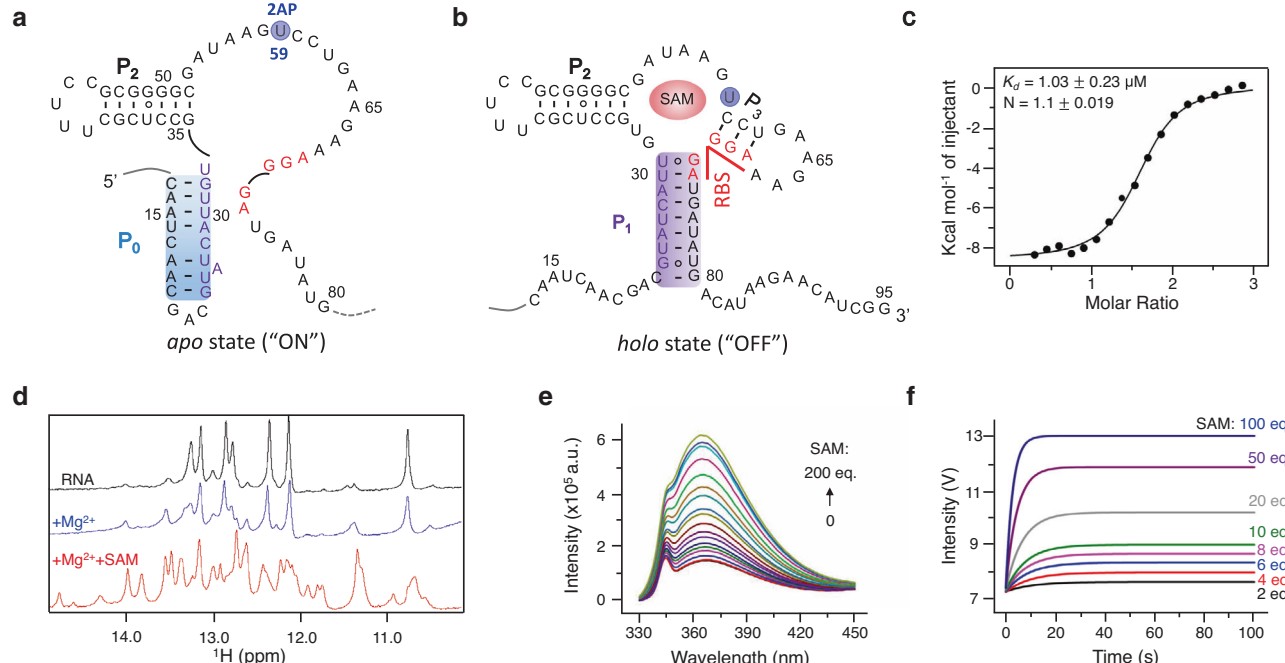

**Fig. 2 | Characteristics of the full-length SAM-VI riboswitch, riboSAM. a** The secondary structure of *apo* contains $P_0$ helix (blue highlighted) in the absence of SAM. **b** The secondary structure of *holo* contains $P_1$ helix (purple highlighted) in the presence of SAM. The nucleotides in RBS are shown in red. The nucleotides shared by helix $P_0$ and $P_1$ are shown in purple. **c** The ITC curve of riboSAM upon the titration of SAM. SAM bound to riboSAM with high affinity ($K_d = 1.03 \pm 0.23\,\mu M$). **d** The NMR spectra of riboSAM (black curve), with 2 mM $Mg^{2+}$ (blue curve) and with 2 mM $Mg^{2+}$ and 0.5 mM SAM (red curve). **e** The steady-state fluorescence spectra of 0.5 µM riboSAM in the presence of 2 mM $Mg^{2+}$ and 0–200-fold SAM. **f** The stopped-flow kinetic curves of 0.5 µM riboSAM after rapidly mixing with 2–100-fold SAM in the presence of 2 mM $Mg^{2+}$. The U at site 59, blue highlighted in (**a**) was replaced by 2AP to collect the data shown in (**e**) and (**f**).

ECs containing transcripts of different lengths are stalled at individual step by omitting certain type(s) of NTPs. In the first step, the bead-DNAs shake with RNA polymerase (RNAP) and a NTP mix missing certain NTP(s) at 37 °C for 15 min. This causes transcription to stall at a specific position, generating ECs containing nascent RNAs. After SPE (solid phase extraction) and extensive washing, the dissociated ECs are removed (blue arrows, Fig. 1a), and the bead-coupled ECs proceed with transcription by addition of another NTP mix (gray arrows, Fig. 1a). Transcription pauses and resumes repeatedly with the addition of less than four types of NTPs at individual step, stalling ECs with transcripts of different length, in which a FRET pair is introduced to the RNA in steps i and j. After step j, the dissociated ECs, containing desthiobiotin-DNA, RNAP, and nascent transcripts with a FRET pair, are immobilized on the slides for subsequent smFRET measurements (Fig. 1b). To achieve single-nucleotide resolution, the following transcription can be elongated by adding only one type of NTP per step (Fig. 1c). In the last step, the ECs are disassembled, and the full-length RNA is isolated from DNA and RNAP, ready for post-transcriptional analysis.

We applied this strategy to study the structural change of nascent riboSAM in ECs, and the transcription was divided into 15 steps (Supplementary Figs. 3a–c). The reagent usages at each step to stall ECs with different transcripts are shown in Supplementary Fig. 3, Tables 1 and 4. Cy3-UTP and Cy5-UTP (Supplementary Figs. 4a, b) were introduced to sites 37 and 74 at steps 6 and 11, respectively. After step 11, the dissociated ECs labeled with a FRET pair, including EC-81 in step 12, EC-86 in step 13 and EC-91 in step 14 were measured by smFRET. Alternatively, elongation from EC-81 to EC-86 can be carried out through steps I to IV, where only one type of NTP was added (Supplementary Fig. 3d). smFRET assays may require 50–200 times dilution of the dissociated ECs when using 10 µM DNA templates (Supplementary Fig. 5). The processive activity of the dissociated ECs decreased with time, and their activity reduced from >70% in 6 h to <20% in 24 h after SPE (Supplementary Figs. 6, 7 and Table 5). Thus, it is highly

recommended to perform smFRET measurements within 6 h after collecting the dissociated ECs.

## Characteristics of the full-length SAM-VI riboswitch

The *Bifidobacterium breve* riboSAM was used in this work, and its secondary structures are ligand-dependent (Fig. 2a, b)[29,30]. In the absence of SAM, the binding pocket is unstructured. The $P_0$ helix (highlighted in blue) stabilizes the *apo* state, exposing the RBS (red nucleotides) to ribosomes and translating downstream genes (Fig. 2a). In contrast, in the presence of SAM, ligand binding disrupts the $P_0$ helix and triggers the $P_1$ helix (highlighted in purple) to form the *holo* state, preventing the binding of ribosomes to RBS and inhibiting translation (Fig. 2b). In the *holo* state, the $P_1$ helix localizes at a three-way junction, with two protruding $P_2$ and $P_3$ helices oriented through the binding pocket. The nucleotides (purple in Fig. 2a, b) overlap in the *apo* and *holo* structures, which give rise to the mutually exclusive folding of the two structures.

We performed ITC, NMR, steady-state fluorescence titration, and stopped-flow kinetics to evaluate the binding between SAM and riboSAM. In the ITC experiment, SAM was titrated to 45 µM riboSAM and exhibited a strong binding affinity with a $K_d$ of $1.03 \pm 0.23$ µM and stoichiometric ratio of 1:1.1 (Fig. 2c and Supplementary Fig. 9), comparable with the reported $K_d$ and stoichiometry[29]. In the NMR spectra of 0.5 mM riboSAM, the number and resolution of peaks in the imino region increase significantly after titration with equivalent SAM at 2 mM $Mg^{2+}$, indicating that SAM binding induced a compact folding of riboSAM (Fig. 2d). We introduced a fluorescent probe, 2-amino-purine (2AP), at site 59 of riboSAM (2AP-riboSAM) to monitor structural changes of riboSAM upon adding SAM by fluorescent assays (Supplementary Figs. 4c, 10, Tables 3 and 6). In the fluorescent spectra, the fluorescence of 2AP-riboSAM increases continuously on adding SAM, and the calculated $K_d$ is $9.650 \pm 0.065$ µM (Fig. 2e and Supplementary Fig. 11), matching the reported $K_d$[30]. In the stopped-flow fluorescence trajectories, sensitive responses of 2AP-riboSAM to 2–100-fold SAM

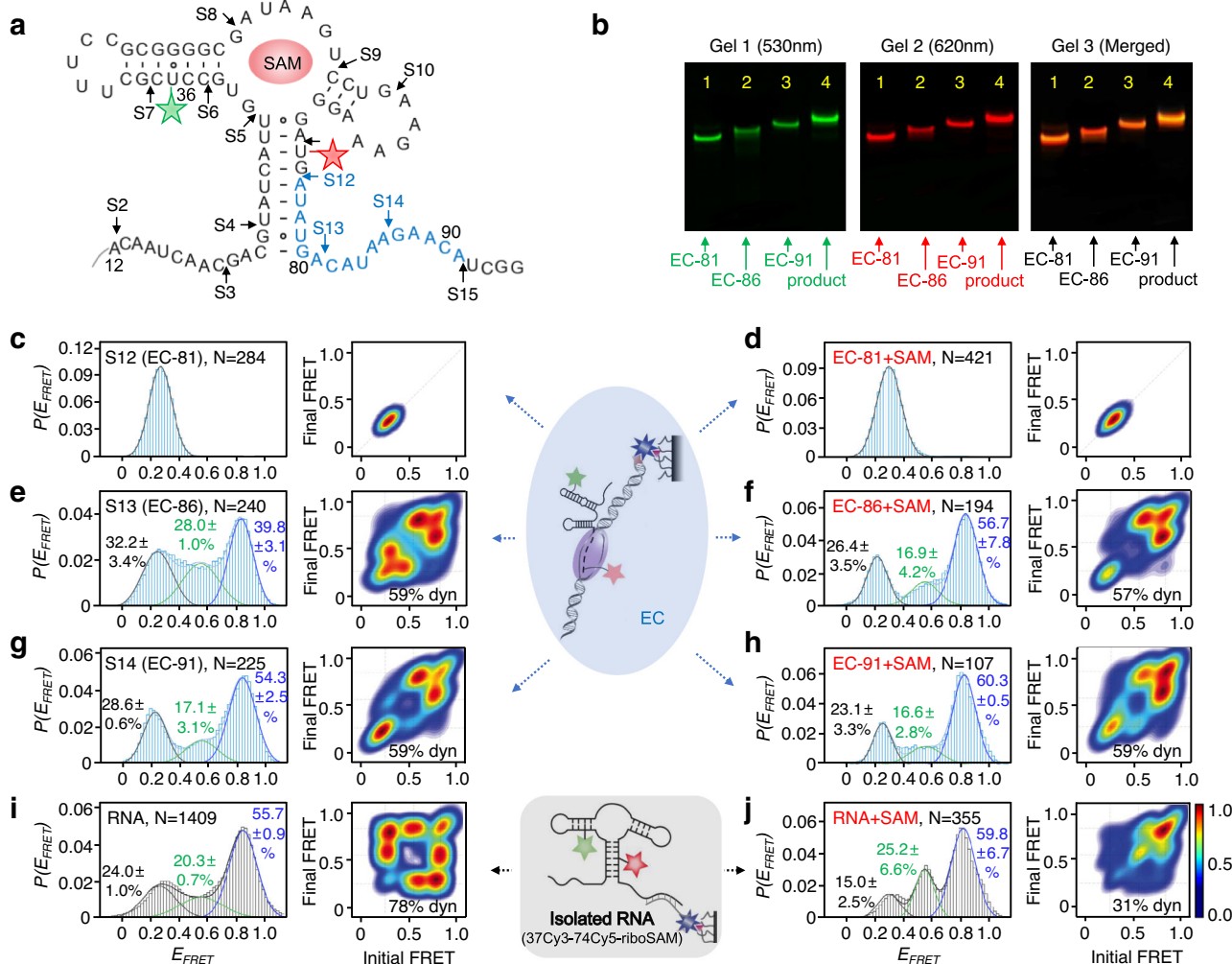

**Fig. 3 | smFRET studies of nascent riboSAM in ECs and isolated riboSAM without and with SAM at 2 mM Mg²⁺.** **a** 15-step reaction was performed to incorporate Cy3 (green star) and Cy5 (red star) into sites 37 and 74 for structural studies of nascent riboSAM in ECs. The ECs dissociated in steps 12, 13, and 14 were used for the co-transcriptional study, and the final product was used for the post-transcriptional study. The nucleotides transcribed in steps 12, 13, and 14 are shown in blue. SAM is shown as a red sphere. **b** Gel shift assays of EC-81, EC-86, EC-91 and the final product (37Cy3-74Cy5-riboSAM). The liquid phase separated by SPE at

steps 12 to 15 was loaded at Lanes 1 to 4. The gels 1 and 2 were irradiated under 530 nm and 620 nm fluorescence, respectively. The gel 3 is the merged image of the gels 1 and 2. The experiment was repeated independently for three times. **c–h** smFRET histograms and transition density plots for EC-81 at 0 and 0.5 mM SAM (**c** and **d**), EC-86 at 0 and 0.5 mM SAM (**e** and **f**), EC-91 at 0 and 0.5 mM SAM (**g** and **h**). **i, j** smFRET histograms and transition density plots for the isolated 37Cy3-74Cy5-riboSAM at 0 and 0.5 mM SAM. Mean values ± s.d. of triplicate experiments are shown for peak percentages.

were detected in less than 20 s, and the calculated association rate constant $K_{on}$ is 5690 ± 83 M⁻¹ s⁻¹ (Fig. 2f and Supplementary Fig. 12). In summary, these experimental results show that the constructed ribo-SAM binds to SAM with high affinity and undergoes structural changes as expectedly.

**Probing structural change of nascent riboSAM in ECs with and without SAM**

We performed a 15-step strategy to incorporate Cy3 (green star) and Cy5 (red star) to sites 37 and 74 of the 95-nt riboSAM (Fig. 3a and Supplementary Fig. 3). The restart sites of riboSAM at each step are marked by arrows and S1, S2, S3, etc. The elongation of nascent RNA at the eluted ECs is consistent with the migration of gel bands (Fig. 3b and Supplementary Fig. 3c). ECs with Cy3/Cy5 were diluted 100-fold and immobilized on the slides for smFRET investigation immediately after SPE (Supplementary Fig. 5d). Excitation of Cy3 or Cy5 dye in an eluted EC showed single photobleaching, which indicates the successful modification of a single Cy3 and Cy5 per RNA in an EC molecule (Supplementary Fig. 5f).

At step 12, the predominant conformation of EC-81 (EC containing 81-nt RNA) adopts a low-FRET structure ($E_{FRET}$ ~ 0.2), indicating riboSAM folds into the *apo* state (Fig. 3c, d). During transcription, the transcriptional bubbles of T7 RNAP include 7–8 nucleotides (nt) of nascent transcripts[41–45]. Thus, for EC-81, it is expected that 73–74 nt participates in the structural folding of its transcript. The insensitivity of EC-81 to SAM indicates that the binding pocket of riboSAM has not formed. This observation is supported by literature showing that the P₁ helix is not folded in a truncated 61-nt riboSAM from *B. angulatum* (corresponding to 73 nt involved in folding herein)[30]. However, the folding patterns of riboSAM undergo noticeable change as extend from EC-81 to EC-86 in step 13 (Fig. 3e, f and Supplementary Fig. 13). Regardless of SAM, three peaks with $E_{FRET}$ ~ 0.2, 0.6, and 0.8 appear, indicating that at least three different riboSAM structures co-exist in EC-86. The high-FRET peak ($E_{FRET}$ ~ 0.8) dominates at EC-86, accounting for about 40% and 57%, respectively, in the absence and presence of 0.5 mM SAM (Fig. 3e, f). The percentages of the high-FRET peak are even higher in step 14 than in step 13 (Fig. 3g, h). The FRET value of the high-FRET peak matches the distance between sites 37 and 74, ~26 Å in the *holo* crystal structure

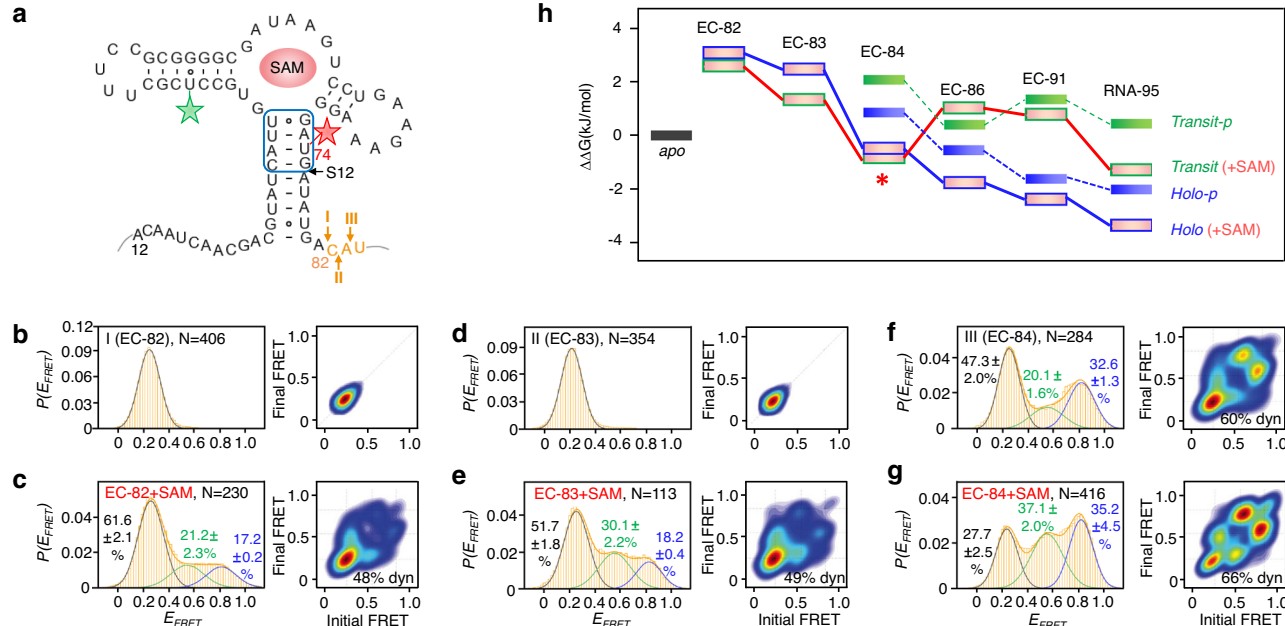

**Fig. 4 | smFRET studies of nascent riboSAM in ECs with and without SAM at single-nucleotide resolution.** **a** The transcription of nucleotides in orange was divided into three steps (I, II, and III), in which only one nucleotide was transcribed per step and generated EC-82, EC-83, and EC-84 used for smFRET measurements. The Cy3-Cy5 FRET pair (green and red stars) was labeled at sites 37 and 74. The base pairs of $P_1$ helix formed in EC-82 at step I are boxed. **b**–**g** smFRET histograms and transition density plots for EC-82 at 0 and 0.5 mM SAM (**b** and **c**), EC-83 at 0 and 0.5 mM SAM (**d** and **e**), and EC-84 at 0 and 0.5 mM SAM (**f** and **g**). The $Mg^{2+}$

concentration used in smFRET was 2 mM. **h** Relative free energy ($\Delta\Delta G$) of *transit-p*, *transit*, *holo-p*, and *holo* structures from EC-82 to EC-91 and the isolated full-length riboSAM (RNA-95) in the absence and presence of SAM. The relative free energy was calculated by using the equation $\Delta\Delta G_{ab} = -RT \ln(P_a/P_b)$, where $R$ and $T$ are the gas constant and absolute temperature, respectively. $P_a$ and $P_b$ are state populations of the observed state and the *apo* state, respectively[32]. Mean values ± s.d. of triplicate experiments are shown for peak percentages.

(PDB ID: 6LAS, Supplementary Fig. 8). Therefore, the high-FRET peak represents the *holo* state. The medium-FRET peak ($E_{FRET}$ ~ 0.6) is an another SAM-bound structure and named *transit* here. Dynamic transition among the three states is observed in steps 13 and 14, however, the addition of SAM increases the stability of SAM-bound structures by either reducing the transition between *apo* and *transit* in step 13 or increasing the rigidity of *holo* in step 14 (Fig. 3e–h).

The final product, 37Cy3-74Cy5-riboSAM, was purified by 12% denaturing PAGE and reversed-phase HPLC (Supplementary Fig. 14 and Table 7). In PAGE, the purified 37Cy3-74Cy5-riboSAM migrates more slowly than its unlabeled counterpart, and is fluorescently visible after excitation at 530 or 620 nm (Supplementary Fig. 14a). In HPLC, fluorescent emission of 37Cy3-74Cy5-riboSAM is observed with excitation wavelengths at 550 and 650 nm, and the retention time is ~3 min longer than the unlabeled counterpart (Supplementary Figs. 14b, c). Both the PAGE and HPLC results are consistent with the fluorescent and hydrophobic characteristics of Cy3 and Cy5 in 37Cy3-74Cy5-riboSAM. The 12-nt biotin-DNA was hybridized to the 3′-end of 37Cy3-74Cy5-riboSAM to immobilize the RNA for smFRET experiments (Supplementary Table 2). After hybridization, the number of nucleotides involved in the structural folding of 95-nt 37Cy3-74Cy5-riboSAM is 83, which is comparable to that of EC-91 in step 14. The isolated 37Cy3-74Cy5-riboSAM and EC-91 share similarities in smFRET, for example, they both fold into at least three states ($E_{FRET}$ of ~ 0.2, 0.6, and 0.8) and *holo* is dominant in the absence or presence of SAM (Fig. 3g–j and Supplementary Figs. 13 and 15). However, the isolated riboSAM and EC-91 display significant differences in dynamics. At 2 mM $Mg^{2+}$, in the absence of SAM, dynamic transition among the three states of the isolated RNA is observed, and the addition of SAM leads to the isolated RNA much more rigid for the *holo* state, and only weak transition among states was detected (Fig. 3i, j). These results differ from EC-91, in which significant dynamic transition between *transit* and *holo* states

is visible in the absence and presence of SAM (Fig. 3g, h). The results show that the isolated RNA and transcripts in ECs behave differently, which is supported by the observation that RNAP and DNA modulate the folding of the preQ$_1$ and fluoride riboswitches[3,16].

## Probing structural switch of nascent riboSAM in ECs at single-nucleotide resolution

When the ECs grow from 81 to 86 nt, smFRET histograms change significantly (Fig. 3c–f). This supports that the five nucleotides transcribed in step 13 play an important role in the structural switch of riboSAM. The conversion between EC-81 and EC-86 was clarified in more details by dividing the transcription of five nucleotides into multiple steps, in which steps I, II and III were extended by a single nucleotide (in orange, Fig. 4a). A significantly different pattern is observed with one nucleotide extension in step I, switching from ligand-independence in EC-81 to SAM-triggered folding in EC-82 (Figs. 3c, d, 4b, c). In step I, in the absence of SAM, only the low-FRET peak (*apo* state) was detected. Adding SAM induces the appearance of two SAM-bound conformations, *holo* ($E_{FRET}$ ~ 0.8) and *transit* ($E_{FRET}$ ~ 0.6) for EC-82. 74−75-nt participates in the structural folding of the transcripts in EC-82, and the existence of the *holo* state signals the formation of the $P_1$ helix by 3–4 bp (blue box, Fig. 4a). The strand exchange between $P_0$ and $P_1$ demonstrates that riboSAM escapes from being ligand-independent. Specifically, in steps I and II, the addition of 0.5 mM SAM induces the occurrence of *holo* and *transit* with SAM-binding characteristics (Fig. 4b–e), and the proportions of the two states increase as transcription proceeds (Fig. 4c, e). Like step I, weak transition is observed between *transit* and *holo* in step II in the presence of SAM. In step III, in the absence of SAM, three states, including *holo* and *transit* occur in EC-84, which differ distinctly from the earlier steps (Fig. 4f and Supplementary Fig. 16). Moreover, in step III, the addition of SAM shifts the predominant state from *apo* to *transit*

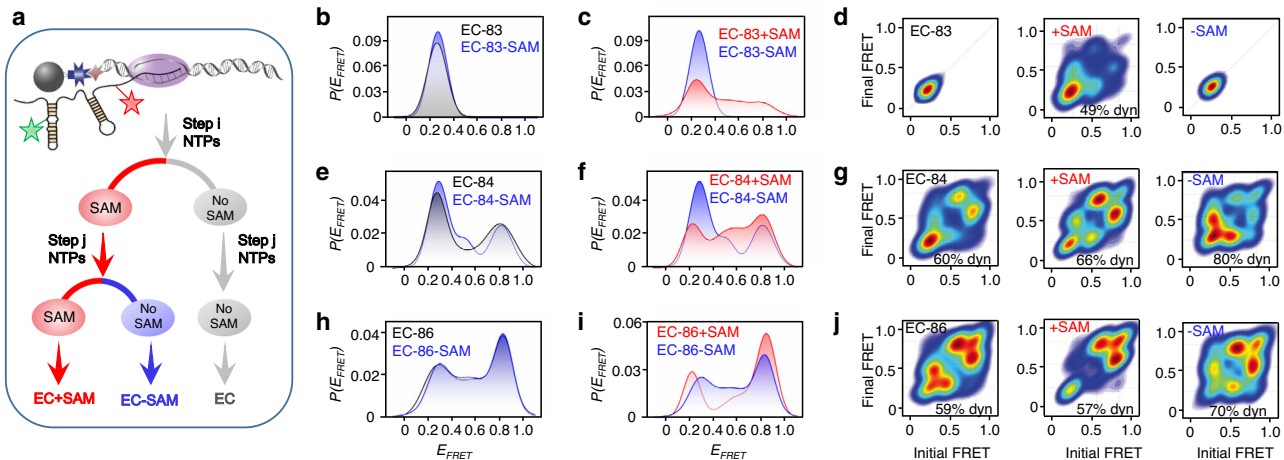

**Fig. 5 | smFRET studies of ECs produced at different SAM conditions.**
**a** Schematic illustration of the additions of SAM in producing EC + SAM, EC-SAM, and EC. EC + SAM and EC are produced in the presence and absence of SAM, respectively. EC-SAM is produced in the same way as EC + SAM except no SAM is added in the last step. **b–j** Superposition of smFRET histograms and transition density plots for EC-83, EC-83-SAM, and EC-83 + SAM (**b–d**), EC-84, EC-84-SAM, and EC-84 + SAM (**e–g**), and EC-86, EC-86-SAM, and EC-86 + SAM (**h–j**).

and causes significant transitions between *transit* and *holo* (Fig. 4g). Our FRET data reveals that the addition of SAM induces the formation of *holo* and *transit* two nucleotides in advance than in the absence of SAM, and both structures are dynamic from EC-82 to EC-84 (Fig. 4c–g). The structures of $E_{FRET}$ ~ 0.6 and 0.8 in the absence of SAM may resemble those in the presence of SAM, except that the binding pockets are not formed until SAM binds. However, the sites labeled with Cy3/Cy5 in 37Cy3-74Cy5-riboSAM are not sensitive to the binding pocket, and thus identifying the formation of the binding pocket based on the FRET values is not possible. To address this issue, we prepared another sample, 16Cy3-62Cy5-riboSAM, placing Cy5 at site 62, close to the binding pocket (Supplementary Fig. 17, Tables 2 and 8). In the absence of SAM, only two peaks ($E_{FRET}$ ~ 0.4 and 0.6) appear. The high-FRET peak ($E_{FRET}$ ~ 0.8) is greatly affected by SAM, which is speculated to be *holo* and *transit* with the structured binding pocket (Supplementary Figs. 17d–k). The low-FRET peak may be the *apo* state, and the medium-FRET peak may represent *transit-p* or *holo-p*, that is, *transit* or *holo* state without the binding pocket.

As transcription proceeds, the proportions and relative free energy of the multiple structures in riboSAM alter significantly (Figs. 3 and 4). In the absence of SAM, *holo-p* ($E_{FRET}$ ~ 0.8) begins to appear in EC-84 at step III, its proportions increase and its relative free energy decreases as transcripts elongate from EC-84 to EC-91 (blue bars, Fig. 4h). The proportions of *transit-p* ($E_{FRET}$ ~ 0.6) increase from ~20% in EC-84 to 28% in EC-86, and then decrease to ~17% in EC-91 (Figs. 4f, 3e, g). Accordingly, the relative free energy of *transit-p* is minimized for EC-86 (green bars, Fig. 4h). In the absence of SAM, the free energy of *holo-p* is lower than *transit-p* from EC-84 to EC-91. In the presence of SAM, the proportions of *holo* increase significantly from ~17% in EC-82 to 60% in EC-91, while the percentages of *transit* increase from ~21% in EC-82 to 37% in EC-84, followed by a drop to ~17% in EC-86 and EC-91 (Figs. 4c, e, g, 3f, h). The relative free energy of *transit* in EC-84 is minimized (marked with red star, Fig. 4h), and *holo* (red bars with blue edges) is lower than *transit* (red bars with green edges) only when the transcripts are longer than 84 nt (Fig. 4h). In summary, the addition of SAM leads to the earlier appearance of *transit* and *holo*, and reduces the relative free energy of these two states except in EC-86. The reduction of relative free energy resulting from the addition of SAM is significant, particularly for EC-84, where the difference in $\Delta\Delta G$ between *transit* and *transit-p* is larger than −2.8 kJ/mol.

To investigate the effect of SAM temporary alteration on riboSAM structures during transcription, we performed the transcription in which SAM was present in the earlier steps and removed from the last step (EC-SAM, Fig. 5a). The ECs generated with and without SAM in the whole transcription are named EC + SAM and EC, respectively (Fig. 5a). The superposition of smFRET histograms between EC-83 and EC-83-SAM, EC-84 and EC-84-SAM, EC-86, and EC-86-SAM suggests that previously adopted *transit* or *holo* structures in the presence of SAM may fold back to *apo* structure once SAM is removed or decreases (Fig. 5b, c, e, f, h, i). Interestingly, the transition patterns among EC, EC-SAM, and EC + SAM are significantly different from each other, and the transition between *apo* and *transit* in EC-SAM is more dynamic than in EC + SAM (Fig. 5g, j).

## Determination of structural change in riboSAM by NMR

To further demonstrate the structural change of riboSAM by the addition of SAM and to obtain the structural information of the identified *transit*, we collected NMR data of riboSAM. It is well-known that RNAs may exist in multiple conformations, and NMR is a powerful method for detecting such conformers[46–48]. However, the strength of NMR is severely limited by peak overlap, especially for long RNAs. Therefore, for RNAs like riboSAM, spectra assignments are expectedly complicated by severe overlap[49,50]. Hence, to greatly simplify the NMR spectra and leave only the peaks of our interest, we prepared two site-specifically labeled samples, [15]N-16U24G-riboSAM and [15]N-25U32G-riboSAM, labeled with [15]N at 16U, 24G (in green circles, Fig. 6a), and 25U, 32G (in orange circles, Fig. 6a), respectively (Supplementary Fig. 18 and Tables 2, 9 and 10). By comparing the [1]H-1D NMR spectra of 0.5 mM [15]N-16U24G-riboSAM at 10–37 °C, peak resolution in the imino region (10–15 ppm) at 25 or 37 °C is higher than at 10 °C (Supplementary Fig. 18c). And 25 or 37 °C was chosen to run the [1]H[15]N-2D HSQC of [15]N-riboSAM.

The reason for us to label a single G and U in one sample for NMR study is that the signals from G or U are located at different [15]N-regions (~140–152 ppm for G and 155–165 ppm for U), which enables to assign peaks from different sites unambiguously[51]. In addition, the distribution of [1]H on NMR spectra can be used to distinguish base-pair types. For example, the [1]H-chemical shifts in standard Watson-Crick (W-C) base pairs are located at ~12–15 ppm, whereas in non-W-C base pairs, such as G-U or G-G mismatches, their signals typically show up with lower chemical shifts[51]. In [15]N-16U24G-riboSAM, a U signal (peak *a′*) and a relatively strong G signal at 12.8 ppm (peak *a*) are observed at 2 mM $Mg^{2+}$ (Fig. 6b), matching 16U and 24G are in standard W-C base pairs at $P_0$ helix in *apo* (Fig. 6a). The weak G signal (peak *u*) is probably from an

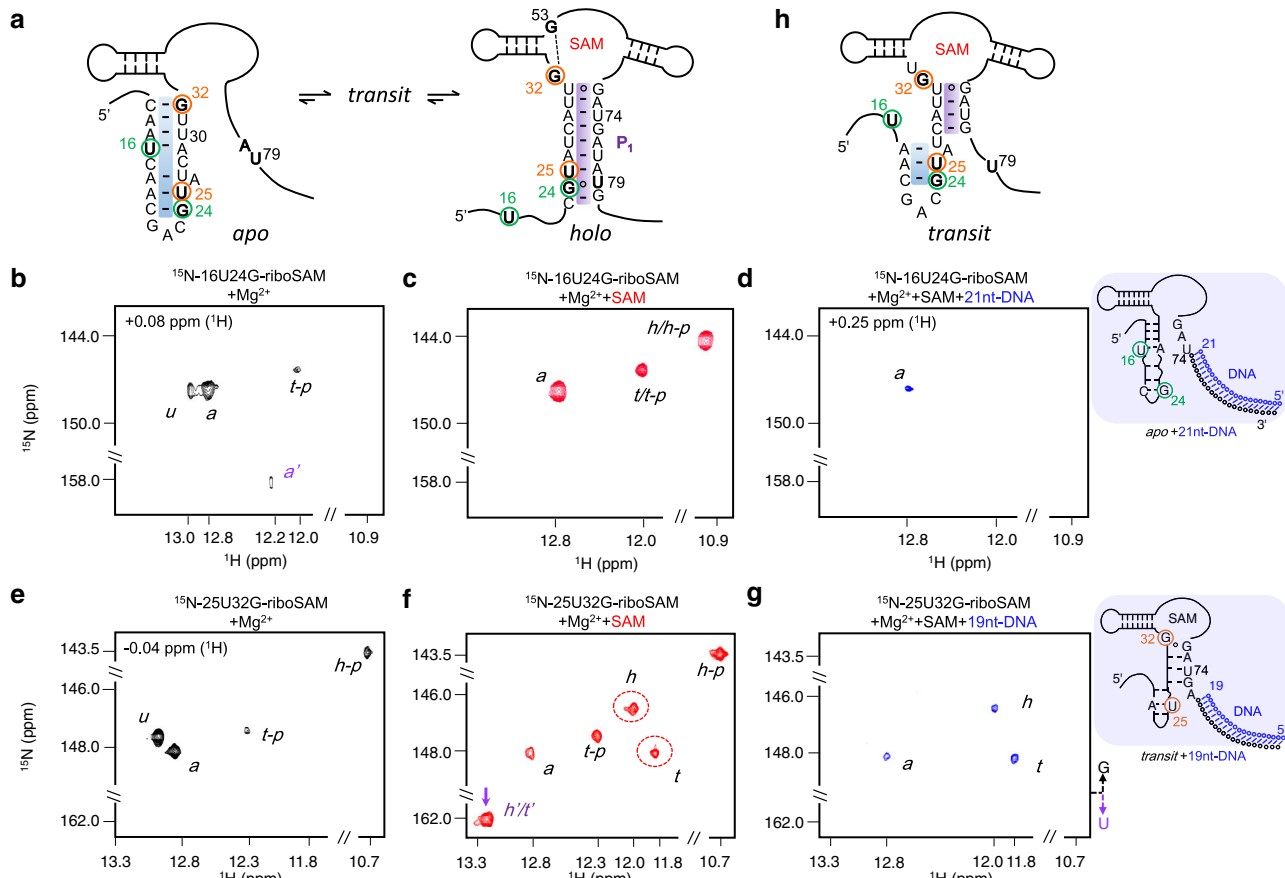

**Fig. 6 | Applications of NMR to determine the structural change of riboSAM.**
**a** The secondary structures of $^{15}$N-riboSAM. The $^{15}$N-labeled sites, 16U and 24G in $^{15}$N-16U24G-riboSAM are shown in green circles. The $^{15}$N-labeled sites, 25U and 32G in $^{15}$N-25U32G-riboSAM are shown in orange circles. **b**–**d** $^{1}$H$^{15}$N-HSQC spectra were recorded for $^{15}$N-16U24G-riboSAM at 2 mM Mg$^{2+}$ in the presence of 0 mM SAM (**b**), 0.5 mM SAM (**c**), 0.5 mM SAM and 21 nt-DNA (**d**). The 21 nt-DNA is shown as blue

circles in (**d**). **e**–**g** $^{1}$H$^{15}$N-HSQC spectra were recorded for $^{15}$N-25U32G-riboSAM at 2 mM Mg$^{2+}$ in the presence of 0 mM SAM (**e**), 0.5 mM SAM (**f**), 0.5 mM SAM, and 19 nt-DNA (**g**). The 19 nt-DNA is shown as blue circles in (**g**). **h** The proposed secondary structure of *transit*. The truncated P$_0$ and P$_1$ in the bifurcated helix are highlighted in blue and purple, respectively.

unfolded or misfolded structure of riboSAM. This is illustrated by the peak getting stronger without Mg$^{2+}$ (Supplementary Fig. 19) but disappearing with Mg$^{2+}$ and SAM (Fig. 6b, c). The addition of SAM also leads to the enhancement of the 24G signals, *t/t-p* and *h/h-p* (Fig. 6c). The new peak *h/h-p* is indicative of the formation of more structure(s), in which 24G is very likely in a non-W-C base pair as illustrated by the much lower $^{1}$H-chemical shift (Fig. 6c). This matches the G-U base pair formed between 24G and 79U in *holo*, and we therefore assigned the new peak as *h/h-p* to represent *holo* and/or *holo-p* (Fig. 6a, c). And the residual peak *t/t-p* should be from *transit* or *transit-p*, and its enhancement with the addition of SAM in Fig. 6c matches the higher proportion of *transit* or *transit-p* with the addition of SAM in smFRET (Figs. 3i, j). And based on the chemical shift, 24G forms a base pair in *transit*. In $^{15}$N-16U24G-riboSAM, the 16U signal (peak *a'*) is much weaker than 24G signals, possibly because of the flexibility and strong chemical exchange of 16U among multiple conformations.

To probe the binding pocket along with the helical switch in riboSAM, we collected the HSQC spectra of $^{15}$N-25U32G-riboSAM upon the addition of Mg$^{2+}$ and SAM. The $^{15}$N-labeled sites, 32G and 25U are located at the binding pocket and helical region (in orange circles, Fig. 6a). At 2 mM Mg$^{2+}$, four peaks from 32G show up at 143–150 ppm (Fig. 6e). The addition of SAM causes a significant change of these signals. Firstly, peak *u* becomes invisible, and we therefore, infer the peak *u* from an unfolded state (Fig. 6f). Peak *a* becomes weaker, matching the lower proportion of *apo* structure in the presence of SAM (Fig. 3i, j). And its $^{1}$H-chemical shift of 12.8 ppm indicates that 32G

forms a W-C base pair in *apo* (Fig. 6a). Peaks *t-p* and *h-p* were inferred from *transit-p* and *holo-p*. Inspection of the crystal structure reveals that 32G N1-H1 imino group and G53 forming a G-G mismatch (Supplementary Fig. 8), which supports the significantly low $^{1}$H-chemical shift of peak *h-p* (-10.7 ppm). The addition of SAM leads to the appearance of new peaks *h* and *t* of 32G (in red circles, Fig. 6f). This indicates the formation of two structures, possibly *transit* and *holo* with reconstructed binding pocket. And they were assigned as *transit* and *holo*, respectively, based on the relative peak percentages in smFRET histograms (Fig. 3j). Moreover, overlapped peaks of 25U show up with the addition of SAM (marked with a purple arrow, Fig. 6f). The use of position-specifically labeled RNAs greatly simplifies the NMR spectra, but peak overlaps are still present, as illustrated by the much larger peak width and decent peak strength of the marked peaks *h'/t'* (Supplementary Fig. 20). Such overlaps are common for RNAs because of the similar chemical environments, such as 25U in *holo* and *transit*. We, therefore, inferred the marked peaks are superposed of 25U from both *holo* and *transit*. And the $^{1}$H-chemical shifts of peaks *h'/t'* are around 13.3 ppm, which indicates that 25U may form standard W-C base pairs in *holo* and *transit*.

Based on the NMR data of $^{15}$N-16U24G- and $^{15}$N-25U32G-riboSAM, we proposed that a bifurcated helix may be formed in *transit* (Fig. 6h). More specifically, *transit* resembles *holo* except the former contains a helix formed by a truncated P$_0$ (blue highlighted) and P$_1$ (purple highlighted, Fig. 6h). To further demonstrate *transit* structure, we designed a labeling scheme with Cy3 (green star) and Cy5 (red star)

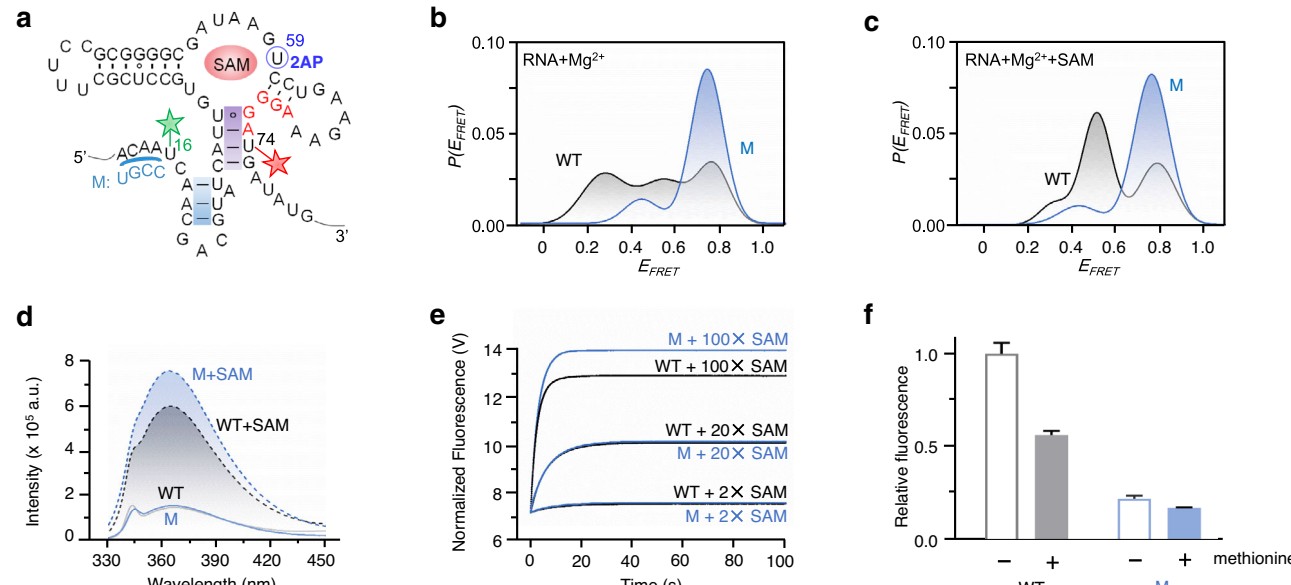

**Fig. 7 | Functional characterization of *transit* in riboSAM. a** The secondary structure of the WT and the mutant of riboSAM, and the mutated nucleotides are in blue. The FRET pair labeled at sites 16 and 74 in 16Cy3-74Cy5-riboSAM is shown by green and red stars. The RBS is shown in red. **b, c** Superposition of smFRET for the WT (in black) and the mutant (in blue) of 16Cy3-74Cy5-riboSAM at 2 mM $Mg^{2+}$ in the presence of 0 (**b**) and 0.5 mM SAM (**c**). **d** The steady-state fluorescence spectra of the WT (in black) and the mutant of riboSAM (in blue) at 2 mM $Mg^{2+}$ in the absence of SAM (solid lines) and in the presence of 0.1 mM SAM (dotted lines). **e** The stopped-flow kinetic curves of 0.5 μM riboSAM (black curves) and its mutant (blue curves) after rapidly mixing with 2-, 20-, or 100-fold SAM at 2 mM $Mg^{2+}$. The U at site 59 in the WT and mutant was replaced by 2AP (blue in (**a**)) to collect the data shown in (**d**) and (**e**). **f** Regulation of RFP expression by the WT or the mutant of riboSAM in *E. coli* in the presence of 0 or 50 mg/L methionine. The data are presented as mean values ± s.d. for three independent experiments. The expression level was adjusted by the internal control, EGFP.

placed at sites 16 and 74 in 16Cy3-74Cy5-riboSAM (Fig. 7a, Supplementary Fig. 21 and Table 11). In the absence of SAM, 16Cy3-74Cy5-riboSAM adopts at least three states with $E_{FRET}$ of about 0.3, 0.6, and 0.8 (Supplementary Fig. 22). $E_{FRET} \sim 0.8$ indicates that sites 16 and 74 are apart by ~20 Å, which should not be generated by the *apo* or *holo* state because the distance between the two sites is unlikely so short with the formation of the $P_1$ helix in *holo* or the $P_0$ helix in *apo* (Fig. 6a). In contrast, the bifurcated helix in *transit* can potentially bring sites 16 and 74 very close, satisfying the short distance required for $E_{FRET} \sim 0.8$. The $E_{FRET}$ of *apo*, *transit* and *holo* obtained from smFRET experiments of 16Cy3-74Cy5-riboSAM agree well with the simulated $E_{FRET}$ by FRET-restrained positioning and screening (FPS) software[52] (Supplementary Fig. 23 and Table 13).

NMR and smFRET are powerful methods for studying conformational change and have been used to study various RNAs. And we here applied these techniques to study riboSAM. The high resolution of NMR helps us to clearly identify the five structures of riboSAM from a single spectrum (Fig. 6f). This strongly supports that at least five structures of riboSAM co-exist in the presence of $Mg^{2+}$ and SAM, as predicted in the previous smFRET results. In addition, the percentages of *apo*, *transit-p*, *transit*, *holo-p*, and *holo* measured in NMR are ~11, 18, 17, 34, and 20% with a $K_d$ of 32G ~27.9 μM (Supplementary Table 12). The results of NMR are consistent with our observation in smFRET, that is, the peak percentages of *apo*, *transit-p* + *transit* and *holo-p* + *holo* are ~15%, 25%, and 60%, respectively (Fig. 3j). To mimic the folding of riboSAM during transcription, we collected the NMR data of the $^{15}N$-labeled samples after hybridization with short DNA strands. Hybridization of 21 nt at 3′-end, the residual 74 nt in $^{15}N$-16U24G-riboSAM is predictable to mimic the folding of ~74 nt transcript in EC-81 (Fig. 6d, Supplementary Fig. 24 and Table 2). And hybridization of 19 nt in $^{15}N$-25U32G-riboSAM mimic the folding of ~76 nt transcript in EC-83 accordingly. In Fig. 6d, peaks from *transit* and *holo* disappear and only peak *a* from *apo* is observable after hybridization, which is consistent with the smFRET results for EC-81 (Fig. 3d). And in Fig. 6g, the

NMR signals from *apo*, *holo* and *transit* can be detected after hybridization, which matches three states co-exist at EC-83 in smFRET (Figs. 6g and 4e). However, the NMR signals *h'/t'* from U25 are not visible, which may be because U25 in the hybridized $^{15}N$-25U32G-riboSAM is too dynamic to be detected by NMR. Our NMR data indicates that *transit* and *holo* may not stabilize unless at least three base pairs are included in $P_1$ (Figs. 3d, 4c, e, 6d, g). The minimum of 3 bp required for forming a helix in guanine riboswitch reported elsewhere[1]. Generally speaking, our NMR and smFRET data are in broad agreement, although different experimental conditions used for NMR *vs.* smFRET, including RNA concentration (~0.5 mM *vs.* 0.5 nM), RNA-to-ligand ratios (~1:1 *vs.* $1:10^6$) and measurement time (6–12 h *vs.* 1–5 min). NMR not only confirms the presence of at least five riboSAM structures, but also supports the decisive switching lengths of riboSAM proposed in smFRET.

**Identifying the function of *transit* by cellular translation assays**
We sought to characterize the *transit* state function by analyzing a mutant that disrupted the $P_0$ helix but preserved base pairing in $P_1$ (Fig. 7a). The disruption of the $P_0$ helix destabilized *apo*, thus increasing the SAM-bound states, however, it was unclear whether this destabilization increased *transit* or *holo*. Therefore, we prepared the mutant labeled with Cy3/Cy5 at sites 16 and 74 (Supplementary Fig. 21 and Table 11) and collected smFRET data at 2 mM $Mg^{2+}$ with and without 0.5 mM SAM (Supplementary Fig. 25). The smFRET histograms of the mutant differ greatly from the wild-type (WT), and >90% of the mutant (M) exists as *transit* or *transit-p* ($E_{FRET} \sim 0.8$) (Fig. 7b, c). The ligand-binding activity of the mutant was evaluated by introducing 2AP to site 59 for fluorescence studies as aforementioned (Fig. 7a, Supplementary Fig. 10 and Table 6). A fluorescence increase was detected for both 2AP-labeled WT and mutant upon SAM addition, and a more drastic increase in the fluorescence spectra and stopped-flow kinetic trajectories are observed for the mutant versus the WT (Fig. 7d, e, Supplementary Figs. 26, 27 and Table 16). The calculated $K_d$ of the

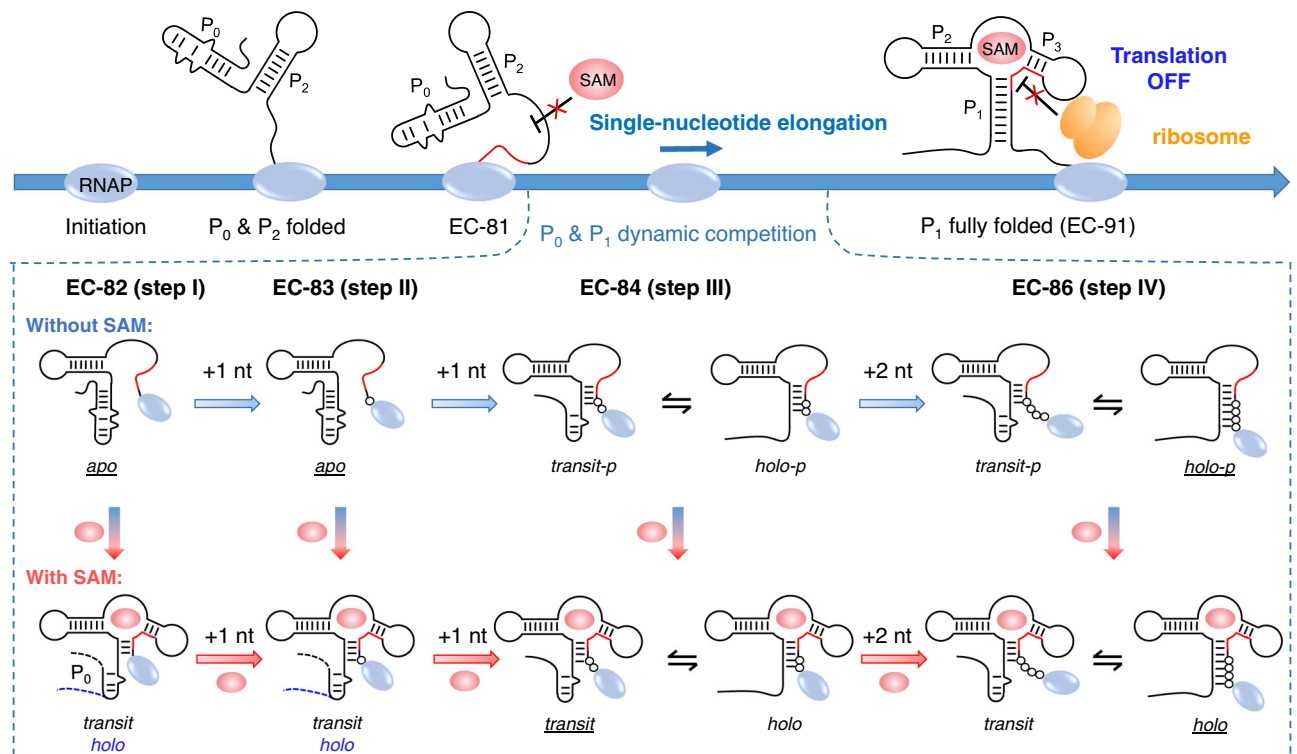

**Fig. 8 | Co-transcriptional conformational landscape of riboSAM in the absence and presence of SAM.** The SAM-sensitive transcription window of riboSAM (EC-82 to EC-86) is shown in the dotted box. The RNAP and SAM are shown as blue and red sphere, respectively. The RBS is shown as a red line. The predominant structures are underlined. The DNA templates, *apo* in the presence of SAM and *apo* in EC-84/EC- 86 in the absence of SAM are not shown in the model. The nucleotides transcribed from EC-82 to EC-86 are shown as black circles. In EC-82, the addition of SAM leads to the appearance of *transit* and *holo*. The structure of *transit* resembles that of *holo*, except in the $P_0$ region. And it first appears at EC-84 and EC-82 in the absence and presence of SAM, respectively.

mutant is comparable with that of the WT ($-11.861 \pm 0.341$ *vs.* $9.650 \pm 0.065$ μM, Supplementary Fig. 26). The results support the SAM-binding characteristics of *transit*.

The regulatory function of the *transit* state in cells was investigated by inserting the WT or mutant upstream of the red fluorescent protein (RFP) in the *pE1K* plasmid (Supplementary Fig. 28). The production of RFP downstream of the WT decreases ~45% after adding 50 μg/mL methionine to *E. coli* at 37 °C (Fig. 7f). This is consistent with the nature of riboSAM, whose RBS is base-paired upon SAM binding to inhibit translation. For the mutant, its predominant state, *transit* or *transit-p*, inhibited the production of RFP significantly regardless of SAM (Fig. 7f), matching the secondary structure of *transit* or *transit-p*, in which the RBS is base-paired and unavailable for ribosome binding (Fig. 7a).

## Discussion

Structural change of nascent RNAs during transcription is related to important biological phenomena, from RNA catalysis to macro-molecular assembly and functions. In this work, our strategy was applied to study the folding of a riboswitch growing at a single nucleotide pace and single-molecule level. Different from analyzing post-transcription of RNAs, which can only examine the final out-comes, the co-transcriptional experiments show distinct advantages for probing intermediate and final states during RNA synthesis like watching a live video, where smFRET trajectories probe distinct nucleotide signatures for conformational rearrangements of indivi-dual growing RNA.

In riboSAM, structural transition occurs at a narrow transcription window, in which the extension of a single nucleotide leads to a drastic change in structural distribution, ligand sensing, and translational decisions. We propose a structural model for nascent riboSAM as transcription proceeds (Fig. 8). The proportions of *apo* decrease as

transcription proceeds, and the presence of SAM reduces the relative free energy of "translation OFF" structures and facilitates functional switching of riboSAM two nucleotides earlier than in the absence of SAM. In general, the structures of riboSAM change more sensitively in the presence of SAM.

*Apo*, the "translation ON" state is stabilized by $P_0$, and therefore, a decrease of *apo* is expected as $P_1$ lengthens. The "translation OFF" states, *transit* (or *transit-p*) and *holo* (or *holo-p*), are formed through unzipping $P_0$ and forming $P_1$, thus the competition of $P_0$ and $P_1$ determines the proportions of the structures. The base pairs formed in *holo* increase from 3–4 bp in EC-82 to 7–8 bp in EC-86, and longer $P_1$ facilitates the stability of *holo*. However, for *transit*, its $P_1$ helix remains 3-4 bp in length during transcription. Therefore, lengthening $P_1$ favors *transit* and *holo* simultaneously at the beginning, but disfavors *transit* after certain transcript lengths. This matches our observations that the predominant state (underlined, Fig. 8) shifts from *apo* in EC-82 to *transit* in EC-84 and then to *holo* in EC-86 in the presence of SAM, whereas it changes more inertly in the absence of SAM, from *apo* in EC-82 to *holo-p* in EC-86 in the absence of SAM. Moreover, in the presence of SAM, the appearance of *transit* or *holo* in EC-82 or hybridized [15]N-25U32G-riboSAM indicates that at least 3 bp in $P_1$ is required for ligand binding in riboSAM. This is supported by previous work on cyclic di-guanosine monophosphate (c-di-GMP), TPP, tetrahydrofolate (THF), and guanine riboswitches, in which ligand-binding also occurs within junctions, and the closing of the $P_1$ helix is critical for maintaining the base of the ligand-binding pocket[1,5,53,54].

It is noteworthy to mention that what we used in the structural study of riboSAM is equilibrated ECs coupled with T7 RNAP. Bacterial RNAP may affect the structures of RNAs differently from T7, and we therefore prepared the equilibrated ECs of riboSAM using *E. coli* RNAP (Supplementary Fig. 29, Tables 14 and 15). As in T7 system, at least

three states are observed in EC-86 and EC-91 while only *apo* is observed in EC-81 in *E. coli* system (Supplementary Figs. 30 and 31). There is no significant difference in smFRET histograms for EC-81 and EC-91 between the two systems. However, the peak proportion of $E_{FRET}$ ~ 0.8 in EC-86 is about 8% and 18% higher in the T7 system than *E. coli* in the absence and presence of SAM, respectively. And more dynamic transition between *transit* and *holo* is observed in T7 system. This may be related to the differences between the two RNAP in size, transcription dynamics, interaction mode with DNA and RNA. This indicates that RNAP plays a part in the ligand-controlled fate of the switching sequence in riboswitches as reported elsewhere[3,16].

Unlike the classical riboswitch model, besides *holo*, another ligand-bound structure, *transit* was identified in this study. The model with at least three states to illustrate the structural basis of riboswitches have been reported for adenine, guanine, fluoride, preQ₁, TPP riboswitches[3,50,54–57]. A distinctive bifurcated helix was determined in *transit* by NMR and smFRET, and such partial helix displacements in *transit* exploit subtle features of RNA folding to enable functions. Based on the relative free energy, *transit* facilitates structural switching from the *apo* to *holo* state by achieving strand replacement with a smaller energetic barrier. The formation of *transit* is possibly critical for inhibiting translation by riboSAM as early as possible, without need to wait until a stable *holo* state is formed. This structural transition may be crucial for the riboswitch to regulate gene expression, especially in sudden alternation of ligand concentration in the environment[58,59]. The RNA structures containing bifurcated helixes were also reported in the co-transcriptional folding of the guanine riboswitch[1]. This may be a vital feature of riboswitch modulation, whereby transient transcriptional windows directly coordinate the sensing of environmental cues and bacterial regulation. In this work, we identified *transit* structure and determined the distinct folding patterns of the SAM-VI riboswitch in the absence and presence of the ligand, which facilities to reveal the structural basis for "switching" in signal transmission and regulatory mechanism of riboswitches, which has remained a mystery since their discovery for more than twenty years ago. In addition, the distinct switching pattern of riboSAM during synthesis can provide a structural basis for antibacterial drug design targeting this riboswitch[60,61]. With the capability of introducing various modifications and obtaining nascent transcripts of desired lengths, our strategy can flexibly cooperate with detection methods to study RNAs when appropriate modifications are incorporated, such as isotopes and spin labels for NMR and EPR, 2'-hydroxyl acylation for SHAPE. As such, our method may have great potential for RNA structural biology in general.

## Methods

### Preparation of ECs of riboSAM using T7 RNAP

The DNA sequences used for the preparation of ECs using T7 RNAP are listed in Supplementary Table 1. The desthiobiotin-DNA templates were prepared by PCR, purified by 10% denaturing PAGE and immobilized on streptavidin-coated agarose beads (Cat. No. SA021010, Smart-Lifesciences, China) as described elsewhere[19]. 10 μL streptavidin-coated beads were added to 1 nmol desthiobiotin-DNAs and incubated at 25 °C for 3 h. The DNA-beads were used as templates to generate ECs in transcription. A detailed description for producing ECs of riboSAM is listed in Supplementary Table 4. In the first step, 10 μM, 100 μL DNA-beads gently rotated with 10 μM T7 RNAP, 1.12 mM ATP, 0.96 mM GTP and 32 μM UTP in the buffer (40 mM Tris-HCl, 100 mM K₂SO₄, 2 mM MgSO₄, 10 mM DTT, pH 8.0) at 37 °C for 15 min. The reaction mixture was filtered by SPE and rinsed at least three times using the buffer (40 mM Tris-HCl, 2 mM MgSO₄, pH 8.0). Except noted, the filtration and bead rinsing were performed between steps. At step 2, the transcription proceeded with the addition of 40 μM ATP, 30 μM CTP and 10 μM UTP in the buffer (40 mM Tris-HCl, 2 mM MgSO₄, pH 8.0) at 37 °C for 10 min. The remaining steps were performed similarly as step 2, but added different NTP mixtures as listed in Supplementary

Table 4. After step 11, the Cy3/Cy5 pair was incorporated into sites 37 and 74 of nascent transcripts. The liquid phase separated from the transcription system by SPE, i.e., EC-81, EC-86, and EC-91 at steps 12, 13, and 14, respectively, were diluted 100 times for smFRET study in 1 h after SPE. And step 13 was replaced by steps I, II, III, and IV to obtain EC-82, EC-83, and EC-84 (Supplementary Table 4).

### Preparation of position-specific labeled RNAs

All RNAs used in this study were in-house prepared. The labeled riboSAM samples, including 37Cy3-74Cy5-riboSAM, 16Cy3-74Cy5-riboSAM/mutant, 16Cy3-62Cy5-riboSAM, ¹⁵N-riboSAM, and 2AP-riboSAM/mutant were prepared by PLOR method as described earlier[19]. The DNA sequences used for RNA syntheses are listed in Supplementary Tables 2 and 3. The biotin-DNA templates were prepared, purified, and then incubated with streptavidin-coated agarose beads to obtain the DNA-bead templates used in PLOR[19]. A detailed description of 37Cy3-74Cy5-riboSAM synthesis is listed in Supplementary Table 7. In the first step, 10 μM, 1 mL DNA-beads gently rotated with 10 μM T7 RNAP, 1.12 mM ATP, 0.96 mM GTP and 32 μM UTP in the buffer (40 mM Tris-HCl, 100 mM K₂SO₄, 6 mM MgSO₄, 10 mM DTT, pH 8.0) at 37 °C for 15 min. The reaction mixture was filtered by SPE and rinsed at least three times in the buffer (40 mM Tris-HCl, 6 mM MgSO₄, pH 8.0). Except noted, the filtration and bead rinsing were performed between steps. In step 2, the transcription proceeded with the addition of 40 μM ATP, 30 μM CTP and 10 μM UTP in the elongation buffer (40 mM Tris-HCl, 6 mM MgSO₄, 10 mM DTT, pH 8.0) at 25 °C for 10 min. The remaining steps were the same as step 2, but added different NTP mixtures as listed in Supplementary Table 7. A higher temperature, 30 °C was used at steps 6 and 11 to increase the efficiencies of introducing Cy3/Cy5 to the RNA. The RNA product at step 12 was then purified by 12% denaturing PAGE and reversed-phase HPLC (RP-HPLC) with C8 column (4.6*250 mm, Cat. No. EXL-122-2546U, Phenomenex Luna, USA) as described earlier[20].

The detailed procedures for the syntheses of 2AP-riboSAM/mutant, 16Cy3-62Cy5-riboSAM, ¹⁵N-riboSAM, and 16Cy3-74Cy5-riboSAM/mutant are listed in Supplementary Tables 6, 8–11. Except noted, the fluorophore-labeled RNAs were purified by 10–15% denaturing PAGE and RP-HPLC. The isotope-labeled RNAs were purified by denaturing PAGE. The purified RNA samples were exchanged to the desired buffer and stored at −20 °C or −80 °C. Except noted, RNA were heated at about 85 °C for 3–5 min, and cooled to room temperature before use.

### Preparation of ECs of riboSAM using *E. coli* RNAP

The DNA and RNA sequences used for the preparation of ECs using *E. coli* RNAP are listed in Supplementary Table 14. *E. coli* RNAP was prepared as described elsewhere[62]. The DNAs were ordered from Sangon Biotech Co., China, and purified by 10% denaturing PAGE. The desthiobiotin-labeled DNA template was annealed with equivalent non-template DNA by heating at 85 °C for 5 min and subsequent cooling to room temperature. The hybridized DNA strands were then immobilized on streptavidin-coated agarose beads to prepare DNA-beads as described earlier. The DNA-beads were used as templates to generate ECs using *E. coli* RNAP as listed in Supplementary Table 15. 5 μM DNA-beads were incubated with equivalent *E. coli* RNAP and 37Cy3-63 nt-RNA at 37 °C for 90 min in the buffer (40 mM Tris-HCl, 2 mM MgSO₄, 10 mM DTT, pH 8.0) to assemble solid-phase EC-63. The EC-63 was rinsed and filtered for at least 3 times by the elongation buffer (40 mM Tris-HCl, 2 mM MgSO₄, pH 8.0) before transcriptional elongation. In the first step of elongation, 30 μM ATP and 20 μM GTP were added to the solid-phase EC-63, and the mixture gently rotated in the elongation buffer at 37 °C for 10 min. Except noted, the reaction mixture was filtered by SPE and rinsed at least three times using elongation buffer before NTP additions. Steps 2 and 3 were performed similarly as step 1, except the added NTPs were 5 μM GTP, 5 μM azide-UTP in step 2, and 15 μM ATP, 5 μM GTP, 10 μM UTP in step 3. At step 4,

the 37Cy3-74azide-carrying EC-beads were incubated with 0.5 mM DBCO-Cy5 (Cat. No. CLK-A130, MCE, USA) in the transcription buffer at 37 °C for 1 h to couple Cy5 at site 74 by click chemistry as reported elsewhere[63]. Steps 5 and 6 were performed similarly as step 1, and the NTPs were added as listed in Supplementary Table 15. The liquid phase separated from the transcription system by SPE, i.e., EC-81, EC-86, and EC-91 at steps 4, 5, and 6, respectively, were diluted 100 times for smFRET study in 1 h after SPE.

### In vitro transcription
The unlabeled riboSAM was prepared by in vitro transcription method. The PCR-generated DNAs were used directly as templates in in vitro transcriptions. 0.02–0.1 μM DNA templates, 0.01–0.05 μM T7 RNAP and 5 mM NTPs were incubated at 37 °C for 3–6 h in the buffer (40 mM HEPES, 10 mM DTT, 28 mM MgCl₂, pH 8.0). The reaction products were purified by 12% denaturing PAGE, then exchanged to the desired buffer and stored at −20 °C before use.

### smFRET assays
0.01 mg/mL streptavidin (Cat. No. 434301, Invitrogen™, USA) in T50 buffer (10 mM Tris, 50 mM NaCl, pH 8.0) was injected into the PEG-passivated flow chamber and incubated for 5 min before being rinsed by T50 buffer. The desthiobiotin-labeled ECs or Cy3/Cy5-labeled riboSAM were immobilized to the coverslip of microscope flow chambers for the smFRET study. The ECs were diluted 50–200 folds and immobilized to the flow chambers. 0.05–0.1 nM Cy3/Cy5-labeled riboSAM samples were immobilized to the flow chambers via hybridized biotin-DNA, which were hybridized with the RNAs in advance (Supplementary Table 2). After incubation for 5 min, the un-immobilized ECs or RNAs were removed from the chambers by the imaging buffer (10 mM Tris-HCl, 50 mM NaCl, 2 mM MgCl₂, 0–0.5 mM SAM, pH 8.0). smFRET data was collected in the imaging buffer, with the addition of 3 mM Trolox (Cat. No. HY-101445, MCE, USA), 5 mM 3,4-dihydroxybenzoic acid (Cat. No. P104382, Shanghai Aladdin Bio-Chem Technology Co., China) and 100 nM protocatechuate dioxygenase to alleviate photobleaching and blinking of Cy3 and Cy5.

smFRET measurements were carried out on an objective-type total internal reflection fluorescence (TRIF) microscopy and an inverted microscope (Eclipse Ti, Nikon, Japan) at 20 °C. The videos with a total time of 100 s were recorded by an EMCCD camera (Andor iXon Ultra 897, UK) with a time resolution of 100 ms. The solid-state 532 nm and 640 nm excitation lasers (OBIS Smart Lasers, Coherent Inc., USA) used digital signals from the camera for modulation. The donor, Cy3 in the FRET pair labeled at ECs or riboSAM was excited by a continuous 532 nm laser. An evanescent field for illumination was generated using a 1.49 NA 1003 oil immersion lens (Apo TIRF, Nikon, Japan). The single-molecule fluorescence trajectories were collected at a frame rate of 10 Hz. Single-molecule videos were preprocessed by the open-sourced software iSMS to extract time-dependent signals[64]. A two-dimensional Gaussian function was used to fit the fluorescent spots on the EMCCD after subtracting the background signal. The software deepFRET was used to identify single-molecule trajectories and photobleaching events, and manual screening was sometimes required to improve the accuracy of data analysis[65]. The single-molecule trajectories with anti-correlated behaviors before photobleaching were used for statistical analysis. The FRET efficiency, $E_{FRET}$ was calculated using the equation:

$$E_{FRET} = I_A/(I_A + I_D).$$

Where $I_A$ and $I_D$ are acceptor and donor fluorescence intensity, respectively. And a Förster radius ($R_0$) of 55 Å was used to calculate an approximate distance between Cy3 and Cy5 by using the equation $E_{FRET} = 1/(1 + (R/R_0)^6)$[66]. The FRET efficiency was idealized using the hidden Markov modeling (HMM) package vbFRET, in which an empirical Bayesian method was used to estimate the FRET states and

the time points of transitions[67]. The transition events were counted based on the state sequence assigned by HMM, and then visualized by the transition density plots (TDP) using the Python module matplotlib.

### NMR spectroscopy
The NMR sample, ¹⁵N-riboSAM was exchanged to the buffer containing 25 mM K₂HPO₄, 50 mM KCl, 10% D₂O (Cat. No. D0200, Innochem, China), pH 6.5. The concentration and volume of ¹⁵N-16U24G-riboSAM in Shigemi tubes were 0.5 mM and 300 μL. The concentration of ¹⁵N-25U32G-riboSAM was 0.15 mM. 2 mM MgCl₂, 0 or equivalent SAM was added to ¹⁵N-riboSAM for collecting the NMR spectra. The two-dimensional ¹H¹⁵N-HSQC spectra were collected at 25 or 37 °C on a Bruker Avance spectrometer at a proton frequency of 600 or 900 MHz and equipped with a triple-resonance cryo-probe (Bruker, Germany). All NMR spectra were processed and analyzed by Topspin 3.6 (Bruker, Germany).

### Isothermal titration calorimetry (ITC) measurements
ITC experiments of riboSAM were performed in the buffer containing 40 mM HEPES, pH 7.5, 100 mM KCl, 10 mM MgCl₂ on a MicroCal ITC 200 (General Electric, USA) at room temperature. 2 μL, 450 μM SAM was injected to 280 μL, 45 μM riboSAM for 18 times with 90 s interval between injections and a 5 μcal/s reference power. The titration between the buffer and riboSAM was performed as background. The ITC data subtracted background was fitted by a one-site binding model using Origin ITC software (OriginLab, USA).

### Steady-state fluorescence spectroscopy
Steady-state fluorescence experiments for 2AP-riboSAM and 2AP-mutant upon titration of SAM were conducted on a FLS 1000 photo-luminescence spectrometer (Edinburgh Instruments Ltd., UK) at room temperature. 0–100 μM SAM was titrated to 0.5 μM, 120 μL 2AP-riboSAM or 2AP-mutant in the buffer containing 10 mM HEPES, pH 7.5, 100 mM KCl, 2 mM MgCl₂. The total volume of added SAM was less than 5 μL in one round of the titration experiment. The fluorescence spectra were recorded from 330 to 450 nm with 308 nm excitation wavelength, 3 nm bandwidth, 1 nm increment, and 2 nm/s scan rate. The fluorescence measurements were repeated three times. The apparent dissociation constant $K_d$ was calculated by the equation:

$$\frac{F - F_0}{F_f - F_0} = \frac{K_d + [\text{SAM}]_{tot} + [\text{RNA}]_{tot} + \{(K_d + [\text{SAM}]_{tot} + [\text{RNA}]_{tot})^2 - 4[\text{SAM}]_{tot}[\text{RNA}]_{tot}\}^{-2}}{2[\text{RNA}]_{tot}}$$

Where $F$, the observed fluorescence; $F_0$, the initial fluorescence; $F_f$, the final fluorescence; $[\text{RNA}]_{tot}$, the concentration of 2AP-riboSAM or 2AP-mutant; $[\text{SAM}]_{tot}$, SAM concentration.

### Stopped-flow fluorescence kinetics
The stopped-flow experiments of 0.5 μM 2AP-labeled riboSAM and its mutant were equally mixed with 1–50 μM SAM at the buffer (10 mM HEPES, 100 mM KCl, 2 mM MgCl₂, pH 7.5) on a SX20 stopped-flow spectrometer (Applied Photophysiscs Ltd., UK). All spectra were collected with 308 nm excitation and a 360 nm long-pass filter at room temperature. The averaged data of over 3 replicates were fitted by Origin 8.5. Observed rate constant $K_{obs}$ was calculated using a single exponential function:

$$F = A \bullet e^{-Kobs.t} + C.$$

Where $F$ represents the change of fluorescence over time ($t$); $C$ represents the initial fluorescence; $A$ is a constant.

### FRET efficiency simulated by FPS
We built the three-dimensional structures of riboSAM by using deep learning ARES network[68,69]. The accessible space of the FRET pair and

the FRET efficiencies were estimated by assuming a quasi-static distribution of inter-dye distances $R_{mp}$. And $R_{mp}$ was obtained from the accessible volume (AV) simulations using FPS software[52]. The ensemble averaged FRET efficiency of the dyes in individual conformation was back-calculated from fluorophore accessible volume (AV) simulations using FPS software. The parameters and results of FPS simulation, including $R_{mp}$, $E_{FRET}$, FRET-pair distance are listed in Supplementary Table 13.

## Cellular translation assays

The sequences of the WT or the mutant of riboSAM were inserted downstream of Enhanced Green Fluorescent Protein (EGFP) and upstream of Red Fluorescent Protein (RFP) at the vector *pE1K*, generating the vector *pE1K-SAM*. DH5α strain with *pE1K-SAM* was grown in the minimal media and then cultured at 37 °C for 10–12 h in the presence of kanamycin, with or without 50 μg/mL methionine. The fluorescence was measured at 510 nm ($\lambda_{ex}$ = 480 nm) for EGFP, and 610 nm ($\lambda_{ex}$ = 580 nm) for RFP on a Cytation™ 5 multi-mode reader (BioTek, USA). The fluorescence measurements were performed at least 3 times, referred by EGFP. The expression level of RFP controlled by the SAM riboswitch was determined by:

$$\text{Expression level} = \frac{R - R_0}{R_c - R_0} \times \frac{E_c - E_0}{E - E_c}$$

Where $R$ and $E$ are the fluorescence of RFP and EGFP from *pE1K-SAM*, $R_c$, and $E_c$ are the fluorescence of RFP and EGFP from *pE1K* vector, $R_0$, and $E_0$ are the fluorescent background of DH5α cells at 610 and 510 nm, respectively.

## Reporting summary

Further information on research design is available in the Nature Portfolio Reporting Summary linked to this article.

# Data availability

The data that support this study are available from the corresponding author upon reasonable request. Source data are provided with this paper.

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

## Acknowledgements

We thank Dr. Zhijun Liu of National Center for Protein Science in Shanghai for assistance in NMR data collection. The work was supported by the National Key Research and Development Program of China [grant no. 2021YFC2100600 and 2021YFA0910300 (Y.L.)] and National Natural Science Foundation of China [grant no. 32071300 and 31872628 (Y.L.)].

## Author contributions

Y.X. and Y.L. designed experiments; Y.X. collected data; J.L., D.C., and X.Z analyzed the FRET data; J.L. ran FPS calculation; D.C. prepared *E. coli* RNAP; J.L., D.C., and L.H. contributed discussions; Y.X. and Y.L. drafted the manuscript and all authors contributed to the revision.

## Competing interests

The authors declare no competing interests.
