## [Peer Review File · Nature Communications]

Observation of structural switch in nascent SAM-VI riboswitch during transcription at single-nucleotide and single-molecule resolutionREVIEWER COMMENTS

Reviewer #1 (Remarks to the Author):

The folding of riboregulators such as riboswitches into their functional structures occurs co-transcriptionally. The kinetics of synthesis and of the process of folding can interfere and thereby modulate the finally adopted RNA structure in the three-dimensional space – and of course different structures will have different functional roles. The present manuscript of Xue et al. is dedicated to such an important question. This is especially of interest as biophysical methods to directly interrogate the folding state during transcription are sparse. The authors suggest using a combination of PLOR (biochemical synthesis of RNAs on solid phase) and smFRET or NMR. This means they synthesize different lengths of the RNA by PLOR, further the method allows incorporation of FRET dyes or isotope labels. By using desthiobiotin instead of biotin as 'attachment molecule', they are able to elute elongation complexes that can be immobilized on smFRET grids and subsequently analyzed. The authors apply this newly developed method to explore the co-transcriptional conformational landscape of the SAM-IV riboswitch. Although this riboswitch acts on translation, the formation of a competing helix structure during transcription determines whether it can bind or not bind the ligand and therefore modulate the association of translation initiation complexes to the riboswitch containing mRNA.

The preparation of different lengths of elongation complexes is successful and through the FRET analysis three different states of the riboswitch can be deduced. These are basically assigned to apo – holo and a third the so-called transit state. Based on these observations in conjunction with their NMR data that also reports on the multistability of the riboswitch under different conditions, the authors devise a co-transcriptional folding route for the SAM-VI riboswitch.

In favor for publication of this manuscript is the fact, that the preparation of a large multitude of elongation complexes is successfully undertaken and that these are good analyzable by smFRET experiments. However, there are some points that in my opinion are critical:

1. The method seems to be very cool, still this it is not a method to explore the folding of RNAs co-transcriptional. It is a method to prepare equilibrated elongation complexes – which is not a bad thing, but is not what is promised in the paper and must not say something about the co-transcriptional folding of the RNA. The important aspect of co-transcriptional folding is kinetics, how fast equilibrate conformations of the RNA in comparison to the speed of transcription. But here transcription is halted, the EC extracted and put in the smFRET measurement device. During this time and treatment any non-equilibrium state that might have been adopted in the course of transcription and then might have get stabilized during elongation of the RNA chain is irretrievably lost.

a. In this respect it would also be necessary to use the cognate or at least the bacterial polymerase instead of T7 polymerase. The kinetic parameters are quite different and therefor have a huge impact on co-transcriptional folding.

2. As transcription and translation control interfere in this riboswitch, modulated by the ligand, it is needed to also perform the transcription in the presence of ligand and then interrogate the system with and subsequently without ligand. This will show if once adopted holo like structures are able to fold back to transit or apo like structures.

3. The NMR analysis of the RNAs falls very short.

a. In the absence of ligand three different conformations are reported (from the G-resonances), one of them is assigned to an unfolded/misfolded state of the RNA but without any rational explanation

b. To more easily compare to the smFRET data the peaks should be integrated and clearly assigned to apo/transit and holo conformations

c. Why is there only a single length analyzed, would it not be necessary to also analyze different length of RNAs by NMR, at least around the proposed decisive length of the RNA, where the switch from apo-to-transit and transit-to-holo occurs?

d. The U resonance can only be there in the apo state, why is the intensity so different to the G resonance in the apo state?

e. The difference between NMR and smFRET data does not arise from different experimental conditions here. It arises from the fact that NMR is much more sensitive to conformational equilibria as smFRET is -at least when the number of states is <8. Therefore the presented NMR data challenges the interpretation of the FRET data in a way, that at least one of the FRET states

represents more than a single conformational state.

Reviewer #2 (Remarks to the Author):

This manuscript develops an smFRET-based method to study folding of the SAM riboswitch. Central to their method is achieving single-nucleotide transcriptional control via nucleotide depletion during solid-phase transcription. Using principles of PLOR, they selectively label nascent RNA at relevant positions to reveal key structural and dynamic characteristics of the SAM binding pocket. At various pause points, nascent RNA (coupled within elongation complexes) is extracted and probed via smFRET in order to understand how the RNA sequence can change structure as it is synthesized.

While this is an interesting approach, it has two severe limitations that make the conclusions reached about the SAM riboswitch mechanism unsupported. The first limitation is that transcription dynamics are not incorporated into the folding conditions. By continually stalling and re-starting transcription, the technique intentionally removes all of the complex dynamics including active transcription and pausing that are characteristic of RNA polymerases, and have been shown to influence RNA folding. The second major limitation is the use of the T7 RNA polymerase which is well known to have different characteristics than bacterial polymerases, again confounding the conclusions. Based on these two limitations it appears that the technique really reports on equilibrium refolded conditions of different intermediate length RNA species. As such, none of the conclusions drawn about cotranscriptional folding are appropriate from this data.

We believe this manuscript is better suited to a methods journal, where perhaps the authors could corroborate their method using one of the riboswitches whose mechanisms have been studied using previous techniques.

Reviewer #3 (Remarks to the Author):

This paper represents a veritable tour-de-force to characterize the dynamic structural ensemble of the SAM-VI riboswitch regulating the translation of SAM synthetase via binding of SAM as a ligand. Using a combination of smFRET, NMR and stopped-flow fluorescence, as well as an assay to isolate fluorescently labeled transcriptional intermediates of the riboswitch, the authors reveal the conformational ensemble of the riboswitch as a function of its transcription, and ligand binding. They identify that the conformational ensemble, and its sensitivity towards SAM changes considerably during transcription. Moreover, they identify an intermediate state between the apo- and holo-form of the riboswitch (a 'transit state'), for which they propose a unique bifurcated structure, involved in lowering the energy of transition between apo- and holo-states. Together, this is a beautiful biophysical study of extensive scope, combining biochemistry, single-molecule and ensemble methods and providing novel insights into riboswitch function and dynamics during its transcription. I thus believe that this study is suitable for publication in Nature Communications, I have however some questions that should be addressed:

1. Regarding the question of induced fit vs. conformational selection: Only because the authors cannot see the holo state in FRET in EC-83 does not mean that it is not there – it could be minor, or too transient for their time resolution. The discrimination between IF and CS thus seems artificial and I don't think they can make the claim of mechanism switching. To really proof an induced fit mechanism, one would need to show that the ligand binds to the apo state first, followed by a conformational transition. The indirect measurements performed here cannot discriminate between the two models, in my opinion, and I would thus strongly suggest to weaken the conclusions regarding the change from IF to CS, unless more data is acquired.

2. Regarding the transit state: The authors assign the observed high FRET state (EFRET = 0.8) to the transit state, containing a bifurcated helix which supposedly results in a structure with the dyes fairly close to each other (Figure 6). While there is a structure for the holo-state and FRET

state assignments can be made directly for this state, the state assignment for the transit state is more arbitrary, and there is no proof for a bifurcated helix (neither from NMR, nor from FRET). Such a proof could be produced by labeling a further nucleotide in the P1 helix, formed upon SAM binding, and using NMR to reveal the base pair formation.

For FRET state assignment, the authors could build a molecular model of the riboswitch containing the bifurcated helix, containing the labels at appropriate positions. To gain a good estimate for FRET states, the FRET positioning and screening (FPS) tool from the Seidel lab can be used (10.1038/NMETH.2222). In fact, this tool could also be helpful to interpret FRET states for apo- and holo-states.

3. Data are lacking statistics, the authors should add errors/uncertainties to their % state assignments, for example, and give number of experimental repeats (not just number of molecules analyzed). This will improve the strength of the conclusions.

4. FRET traces in the SI rarely show the intermediate FRET state (eg. in Fig S15e). Is this state only found in a subset of traces, or is it stochastically visited?

5. The authors attribute different conditions to the different observed state distributions between NMR and FRET experiments. Could the authors estimate, based on K_d , what percentage of state occupancy is expected and match this with the observation?

6. I got confused about the assignment of the sites in supplementary figure 3: Step 12 supposedly produces HC-81, the first molecule evaluated by smFRET (Fig. S3b). In the scheme S3a, the blue arrow labeled S12 however points at position 75. Position 81 is marked by S13, etc. The authors should double check their labeling in either S3a or S3b, one if incorrect.

7. The English requires some further careful editing, in particular in the beginning of the paper.

RESPONSE TO REVIEWERS' COMMENTS

Reviewer #1 (Remarks to the Author):

The folding of riboregulators such as riboswitches into their functional structures occurs co-transcriptionally. The kinetics of synthesis and of the process of folding can interfere and thereby modulate the finally adopted RNA structure in the three-dimensional space – and of course different structures will have different functional roles. The present manuscript of Xue et al. is dedicated to such an important question. This is especially of interest as biophysical methods to directly interrogate the folding state during transcription are sparse. The authors suggest using a combination of PLOR (biochemical synthesis of RNAs on solid phase) and smFRET or NMR. This means they synthesize different lengths of the RNA by PLOR, further the method allows incorporation of FRET dyes or isotope labels. By using desthiobiotin instead of biotin as 'attachment molecule', they are able to elute elongation complexes that can be immobilized on smFRET grids and subsequently analyzed. The authors apply this newly developed method to explore the co-transcriptional conformational landscape of the SAM-IV riboswitch. Although this riboswitch acts on translation, the formation of a competing helix structure during transcription determines whether it can bind or not bind the ligand and therefore modulate the association of translation initiation complexes to the riboswitch containing mRNA.

The preparation of different lengths of elongation complexes is successful and through the FRET analysis three different states of the riboswitch can be deduced. These are basically assigned to apo – holo and a third the so-called transit state. Based on these observations in conjunction with their NMR data that also reports on the multistability of the riboswitch under different conditions, the authors devise a co-transcriptional folding route for the SAM-VI riboswitch.

In favor for publication of this manuscript is the fact, that the preparation of a large multitude of elongation complexes is successfully undertaken and that these are good analyzable by smFRET experiments. However, there are some points that in my opinion are critical:

- 1. The method seems to be very cool, still this it is not a method to explore the folding of RNAs co-transcriptional. It is a method to prepare equilibrated elongation complexes – which is not a bad thing, but is not what is promised in the paper and must not say something about the co-transcriptional folding of the RNA. The important aspect of co-transcriptional folding is kinetics, how fast equilibrate conformations of the RNA in*

comparison to the speed of transcription. But here transcription is halted, the EC extracted and put in the smFRET measurement device. During this time and treatment any non-equilibrium state that might have been adopted in the course of transcription and then might have get stabilized during elongation of the RNA chain is irretrievably lost.

Response:

We thank the reviewer for describing our method “very cool” and bringing “co-transcriptional folding” to our attention. We agree with the reviewer and we have revised “co-transcriptional folding” to “structural change/switch during transcription”, “structures of nascent RNA”, or “structure changes as transcription proceeds”, *etc.* Furthermore, we have added a statement to avoid the confusion in the revised manuscript accordingly as “It is noteworthy to mention that what we used in the structural study of riboSAM is equilibrated ECs coupled with T7 RNAP” (highlighted at page 18).

a. In this respect it would also be necessary to use the cognate or at least the bacterial polymerase instead of T7 polymerase. The kinetic parameters are quite different and therefor have a huge impact on co-transcriptional folding.

Response:

We thank the reviewer for the great comment! We prepared the equilibrated ECs of riboSAM using *E. coli* RNAP. And we observed differences in the smFRET histograms and transition plots (Supplementary Figs. 29-31 and Tables 14-15).

We have added the new data to the revised manuscript as “Bacterial RNAP may affect the structures of RNAs differently from T7, and we therefore prepared the equilibrated ECs of riboSAM using *E. coli* RNAP (Supplementary Fig. 29, Tables 14 and 15). As in T7 system, at least three states were observed in EC-86 or EC-91 while only *apo* was observed in EC-81 in *E. coli* system (Supplementary Figs. 30 and 31). There is no significant difference in smFRET histograms for EC-81 and EC-91 between the two systems. However, the peak proportions of $E_{\text{FRET}} 0.8$ are obviously higher in the T7 system than *E. coli* for EC-86. And more dynamic transitions, especially between *transit* and *holo* were observed in T7 system, too. This indicates that RNAP plays a part in the ligand-controlled fate of the switching sequence in riboswitches as reported elsewhere^{3,16}” (highlighted at page 18).

2. As transcription and translation control interfere in this riboswitch, modulated by the ligand, it is needed to also perform the transcription in the presence of ligand and then

interrogate the system with and subsequently without ligand. This will show if once adopted holo like structures are able to fold back to transit or apo like structures.

Response:

We thank the reviewer for the great comment! We collected smFRET data of the ECs produced when SAM was present except the last step. And the superposition of smFRET for EC (generated in the absence of SAM) and EC-SAM (generated in the presence of SAM except the last step) indicates that adopted *holo* and *transit* may fold back to *apo* after removal of SAM (Figure 5).

We have added the new data in the revised manuscript as “To investigate the effect of SAM temporary alteration on riboSAM structures during transcription, we performed the transcription in which SAM was present in the earlier steps and removed from the last step (EC-SAM, Fig. 5a). The ECs generated with and without SAM in the whole transcription are named EC+SAM and EC, respectively (Fig. 5a). The superposition of smFRET histograms between EC-83 and EC-83-SAM, EC-84 and EC-84-SAM, EC-86 and EC-86-SAM suggests that previously adopted *transit* or *holo* structures in the presence of SAM may fold back to *apo* structure once environmental SAM is removed or decreases (Figs. 5b, c, e, f, h and i). Interestingly, the transition patterns among EC, EC-SAM and EC+SAM are significantly different from each other, and the transitions between *apo* and *transit* in EC-SAM is more dynamic than in EC+SAM (Figs. 5g and j)” (highlighted at pages 11 and 12).

3. The NMR analysis of the RNAs falls very short.

a. In the absence of ligand three different conformations are reported (from the G-resonances), one of them is assigned to an unfolded/misfolded state of the RNA but without any rational explanation.

Response:

We thank the review for the comment. We have expanded NMR analysis (highlighted at pages 12-16). Specifically, we have added a figure (Supplementary Fig. 19) to explain the assignment of an unfolded/misfolded state in the revised manuscript as “The weak G signal (peak *u*) is probably from an unfolded or misfolded structure of riboSAM, as illustrated by the same peak also appearing without Mg²⁺ (Supplementary Fig. 19) but disappearing with Mg²⁺ and SAM (Figs. 6b, c)” (highlighted at page 13).

b. To more easily compare to the smFRET data the peaks should be integrated and clearly

assigned to apo/transit and holo conformations

Response:

We thank the reviewer for the comment. We have integrated NMR peaks (Supplementary Table 12), and made some NMR assignments based on the proportions of smFRET peaks.

We have revised the manuscript accordingly as “And the residual peak *t/t-p* should be from *transit* or *transit-p*, and its enhancement with the addition of SAM in Fig. 6c matches the higher proportion of *transit* or *transit-p* with the addition of SAM in smFRET (Figs. 3i, j)”, “Peak *a* becomes weaker, matching the lower proportion of *apo* structure in the presence of SAM (Figs. 3i and 3j)” (highlighted at page 13), “Peaks *t-p* and *h-p* enhance greatly with SAM (Fig. 6f), and were inferred from *transit-p* and *holo-p* with consideration of their relative peak percentages in smFRET histograms (Fig. 3i) ... The addition of SAM leads to the appearance of new peaks *h* and *t* of 32G (in red circles, Fig. 6f). This indicates the formation of two new structures, possibly *transit* and *holo* with reconstructed binding pocket, which were also assigned based on the relative peak percentages in smFRET histograms” (highlighted at pages 13-14), and “In addition, the percentages of *apo*, *transit-p*, *transit*, *holo-p* and *holo* measured in NMR are ~11, 18, 17, 34 and 20% with a K_d of 32G ~27.9 μ M (Supplementary Table 12). The results of NMR are consistent with our observation in smFRET, that is, the peak percentages of *apo*, *transit-p+transit* and *holo-p+holo* are ~15, 25 and 60% respectively (Fig. 3g)” (highlighted at page 15).

c. Why is there only a single length analyzed, would it not be necessary to also analyze different length of RNAs by NMR, at least around the proposed decisive length of the RNA, where the switch from apo-to-transit and transit-to-holo occurs?

Response:

We thank the reviewer for the great comment. We synthesized a new NMR sample, ^{15}N -25U32G-riboSAM, with ^{15}N labeled at sites 25U and 32G (Fig. 6a and Supplementary Table 10), and hybridized different lengths at 3'-end of ^{15}N -25U32G- and ^{15}N -16U24G-riboSAM to mimic RNA folding at decisive length (Figs. 6d, g and Supplementary Fig. 24). And their NMR results are consistent with the switching length of riboSAM from smFRET data.

We have added the new data in the revised manuscript as “To mimic the folding of riboSAM during transcription, we collected the NMR data of the ^{15}N -labeled samples after hybridization with short DNA strands. Hybridization of 21 nt at 3'-end,

the residual 74 nt in ¹⁵N-16U24G-riboSAM is predictable to mimic the folding of ~74 nt transcript in EC-81 (Supplementary Fig. 24 and Table 2). And hybridization of 19 nt in ¹⁵N-25U32G-riboSAM mimic the folding of ~76 nt transcript in EC-83 accordingly. In Fig. 6d, peaks from *transit* and *holo* disappear and only peak *a* from *apo* is observable after hybridization, which is consistent with the smFRET results for EC-81 (Fig. 3d). And in Fig. 6g, the NMR signals from *apo*, *holo* and *transit* can be detected after hybridization, which matches three states co-exist at EC-83 in smFRET (Fig. 4e). This indicates that *transit* and *holo* may not stabilize unless at least three base pairs are included in P₁ (Figs. 3d, 4e, 6d and g)” (highlighted at page 15).

d. The U resonance can only be there in the apo state, why is the intensity so different to the G resonance in the apo state?

Response:

We thank the reviewer for the comment. We have added an explanation in the revised manuscript as “In ¹⁵N-16U24G-riboSAM, the 16U signal (peak *a*) is much weaker than 24G signals, possibly because of the flexibility and strong chemical exchange of 16U among multiple conformations” (highlighted at page 13).

e. The difference between NMR and smFRET data does not arise from different experimental conditions here. It arises from the fact that NMR is much more sensitive to conformational equilibria as smFRET is -at least when the number of states is <8. Therefore the presented NMR data challenges the interpretation of the FRET data in a way, that at least one of the FRET states represents more than a single conformational state.

Response:

We thank the reviewer for the great comment! We agree with the reviewer that NMR is more sensitive to equilibrated conformations, as illustrated from our new collected NMR data that five states of riboSAM can be observed directly from a single spectrum (Fig. 6f).

We have revised the manuscript accordingly by adding “NMR and smFRET are powerful methods for studying conformation change and have been used to study various RNAs. And we here apply these techniques to study riboSAM for the first time. The high resolution of NMR helps us to clearly identify the five structures of riboSAM from a single spectrum (Fig. 6f). This strongly supports that at least five structures of riboSAM co-exist in the presence of Mg²⁺ and SAM, as predicted in the previous smFRET results ... NMR not only confirms the presence of at least five

riboSAM structures, but also supports the decisive switching lengths of riboSAM proposed in smFRET” (highlighted at pages 15 and 16).

Reviewer #2 (Remarks to the Author):

This manuscript develops an smFRET-based method to study folding of the SAM riboswitch. Central to their method is achieving single-nucleotide transcriptional control via nucleotide depletion during solid-phase transcription. Using principles of PLOR, they selectively label nascent RNA at relevant positions to reveal key structural and dynamic characteristics of the SAM binding pocket. At various pause points, nascent RNA (coupled within elongation complexes) is extracted and probed via smFRET in order to understand how the RNA sequence can change structure as it is synthesized.

While this is an interesting approach, it has two severe limitations that make the conclusions reached about the SAM riboswitch mechanism unsupported. The first limitation is that transcription dynamics are not incorporated into the folding conditions. By continually stalling and re-starting transcription, the technique intentionally removes all of the complex dynamics including active transcription and pausing that are characteristic of RNA polymerases, and have been shown to influence RNA folding.

Response:

We thank the reviewer for the comments and hope that our answers will convince him of the suitability of our paper for Nature Communications. We may mislead the reviewer by using “co-transcriptional folding” incorrectly in the manuscript. Actually, what we focus is the structural switch of nascent RNAs in ECs instead of transcriptional dynamics. To strength it, we prepared a new ¹⁵N-labeled sample and nascent riboSAM generated at different ligand conditions to expand the NMR and smFRET analysis (Figs. 5, 6d-g and Supplementary Table 10). We also built structural models with the FRET-pair by deep learning program and FPS software (Supplementary Fig. 23 and Table 13). And to avoid the confusion, we removed “co-transcriptional folding” from the text, changed the title from “Observation of co-transcriptional folding in single SAM-VI riboswitch molecule at single-nucleotide resolution” to “Observation of structural switch in a growing SAM-VI riboswitch at single-nucleotide and single-molecule resolution”, rewrote the abstract and some conclusions in the text, and added a statement in the text to emphasize our usage of equilibrated ECs in the revised manuscript as “It is noteworthy to mention that what we used in the structural study of riboSAM is equilibrated ECs coupled with T7 RNAP” (highlighted at page 18).

The second major limitation is the use of the T7 RNA polymerase which is well known to

have different characteristics than bacterial polymerases, again confounding the conclusions. Based on these two limitations it appears that the technique really reports on equilibrium refolded conditions of different intermediate length RNA species. As such, none of the conclusions drawn about cotranscriptional folding are appropriate from this data.

Response:

We thank the reviewer for the comment. We agree with the reviewer that T7 RNAP affect RNA structures differently from bacterial RNAP and our technique is used to obtain the equilibrated ECs containing transcripts of different lengths in this work. Therefore, we also prepared equilibrated transcription complexes using *E. coli* RNAP and observed the structural switch of riboSAM in *E. coli* (Supplementary Figs. 29-31, Tables 14 and 15).

We have added the new data to the revised manuscript as “It is noteworthy to mention that what we used in the structural study of riboSAM is equilibrated ECs coupled with T7 RNAP. Bacterial RNAP may affect the structures of RNAs differently from T7, and we therefore prepared the equilibrated ECs of riboSAM using *E. coli* RNAP (Supplementary Fig. 29, Tables 14 and 15). As in T7 system, at least three states were observed in EC-86 or EC-91 while only *apo* was observed in EC-81 in *E. coli* system (Supplementary Figs. 30 and 31). There is no significant difference in smFRET histograms for EC-81 and EC-91 between the two systems. However, the peak proportions of $E_{\text{FRET}} 0.8$ are obviously higher in the T7 system than *E. coli* for EC-86. And more dynamic transitions, especially between *transit* and *holo* were observed in T7 system, too. This indicates that RNAP plays a part in the ligand-controlled fate of the switching sequence in riboswitches as reported elsewhere” (highlighted at page 18).

We believe this manuscript is better suited to a methods journal, where perhaps the authors could corroborate their method using one of the riboswitches whose mechanisms have been studied using previous techniques.

Response:

We respectfully disagree on the Reviewer 2’s statement. Our method has great potential for RNA structural study in general, including the riboswitches whose mechanisms have been studied using previous techniques as the reviewer mentioned. However, we did not intend to make an impression that we only focused on developing a new method. What we mean to emphasize is not only the development of the method, but also the determination of the structural switch of SAM-VI

riboswitch, which was discovered in 2018 and has thus far been reported with a *holo* crystal. And no NMR or smFRET data has been reported for this RNA. It is well-known that the detailed structural switch of riboswitches are required to solve long-standing questions in riboswitches: “Do distinct conformations exist in riboswitches as they are synthesized, how are the conformation switch controlled by ligand binding, and how are such structural changes transmitted to the regulation of gene expression?”. To answer these important questions, the detailed structural knowledge of more riboswitches is required. As such, we aim to demonstrate the potential of our developed method and to open new horizons for riboswitch biology by a relatively new, less-documented RNA instead of a RNA that has been studied widely.

Reviewer #3 (Remarks to the Author):

This paper represents a veritable tour-de-force to characterize the dynamic structural ensemble of the SAM-VI riboswitch regulating the translation of SAM synthetase via binding of SAM as a ligand. Using a combination of smFRET, NMR and stopped-flow fluorescence, as well as an assay to isolate fluorescently labeled transcriptional intermediates of the riboswitch, the authors reveal the conformational ensemble of the riboswitch as a function of its transcription, and ligand binding. They identify that the conformational ensemble, and its sensitivity towards SAM changes considerably during transcription. Moreover, they identify an intermediate state between the apo- and holo-form of the riboswitch (a 'transit state'), for which they propose a unique bifurcated structure, involved in lowering the energy of transition between apo- and holo-states. Together, this is a beautiful biophysical study of extensive scope, combining biochemistry, single-molecule and ensemble methods and providing novel insights into riboswitch function and dynamics during its transcription. I thus believe that this study is suitable for publication in Nature Communications, I have however some questions that should be addressed:

1. Regarding the question of induced fit vs. conformational selection: Only because the authors cannot see the holo state in FRET in EC-83 does not mean that it is not there – it could be minor, or too transient for their time resolution. The discrimination between IF and CS thus seems artificial and I don't think they can make the claim of mechanism switching. To really proof an induced fit mechanism, one would need to show that the ligand binds to the apo state first, followed by a conformational transition. The indirect measurements performed here cannot discriminate between the two models, in my opinion, and I would thus strongly suggest to weaken the conclusions regarding the change from IF to CS, unless more data is acquired.

Response:

We thank the reviewer very much for the enthusiasm he expresses for our technology. We agree with the reviewer that we may miss the *holo* state due to too low proportion or too transient, and we have removed the content of switching from IF to CS from discussions in the text and revised Figure 8 (Figure 7 in the original version) accordingly.

2. Regarding the transit state: The authors assign the observed high FRET state (EFRET = 0.8) to the transit state, containing a bifurcated helix which supposedly results in a structure with

the dyes fairly close to each other (Figure 6). While there is a structure for the holo-state and FRET state assignments can be made directly for this state, the state assignment for the transit state is more arbitrary, and there is no proof for a bifurcated helix (neither from NMR, nor from FRET).

Such a proof could be produced by labeling a further nucleotide in the P1 helix, formed upon SAM binding, and using NMR to reveal the base pair formation.

Response:

We thank the reviewer for the great comment. We synthesized a new sample, ^{15}N -25U32G-riboSAM, with ^{15}N -labeled at sites 25U in the P1 helix and 32G in the binding pocket to provide more NMR support for *transit* structure (Figs. 6e-g and Supplementary Table 10).

We have added the new data in the revised manuscript as “To probe binding pocket along with helical switch in riboSAM, we collected the HSQC spectra of ^{15}N -25U32G-riboSAM upon the addition of Mg^{2+} and SAM. The ^{15}N -labeled sites, 32G and 25U are located at the binding pocket and helical region (in orange circles, Fig. 6a). At 2 mM Mg^{2+} , four peaks from 32G show up at 143-150 ppm (Fig. 6e). The addition of SAM causes significant change of these signals. Firstly, peak *u* becomes invisible, and we therefore, infer the peak *u* from an unfolded state (Fig. 6f). Peak *a* becomes weaker, matching the lower proportion of *apo* structure in the presence of SAM (Figs. 3i and 3j). And its ^1H -chemical shift of 12.8 ppm indicates that 32G forms a W-C base pair in *apo* (Fig. 6a). Peaks *t-p* and *h-p* enhance greatly with SAM (Fig. 6f), and were inferred from *transit-p* and *holo-p* with consideration of their relative peak percentages in smFRET histograms (Fig. 3i). Inspection of the crystal structure reveals that 32G N1-H1 imino group and G53 forming a G-G mismatch (Supplementary Fig. 8), which supports the significantly low ^1H -chemical shift of peak *h-p* (~ 10.7 ppm). The addition of SAM leads to the appearance of new peaks *h* and *t* of 32G (in red circles, Fig. 6f). This indicates the formation of two new structures, possibly *transit* and *holo* with reconstructed binding pocket, which were assigned also based on the relative peak percentages in smFRET histograms (Fig. 3j). Moreover, imino peaks of 25U show up with the addition of SAM (marked with a purple arrow, Fig. 6f). The use of position-specifically labeled RNAs greatly simplifies the NMR spectra, but peak overlaps are still present, as illustrated by the much larger peak width and decent peak strength of the marked peaks *h'/t'* (Supplementary Fig. 20). Such overlaps are common for RNAs because of the similar chemical environments in multiple conformations, such as 25U in *holo* and *transit*. We therefore inferred the marked peaks are superposed of 25U from both *holo* and

transit. And the ¹H-chemical shifts of peaks *h'/t'* are around 13.3 ppm, which indicates that 25U may form standard W-C base pairs in *holo* and *transit*" (highlighted at pages 13 and 14).

For FRET state assignment, the authors could build a molecular model of the riboswitch containing the bifurcated helix, containing the labels at appropriate positions. To gain a good estimate for FRET states, the FRET positioning and screening (FPS) tool from the Seidel lab can be used (10.1038/NMETH.2222). In fact, this tool could also be helpful to interpret FRET states for apo- and holo-states.

Response:

We thank the reviewer for the great comment! We built structural models of *apo*, *transit* and *holo* by deep learning program and back-calculation of FRET efficiencies of the structures by FPS software (Supplementary Fig. 23 and Table 13). And we revised the manuscript accordingly as "The E_{FRET} of *apo*, *transit* and *holo* obtained from smFRET experiments all agree well with the simulated E_{FRET} by FRET-restrained positioning and screening (FPS) software⁵²" (highlighted at page 15).

3. Data are lacking statistics, the authors should add errors/uncertainties to their % state assignments, for example, and give number of experimental repeats (not just number of molecules analyzed). This will improve the strength of the conclusions.

Response:

We thank the reviewer for the comment. In the revised manuscript, we have added errors/uncertainties of peak proportions to Figs. 3, 4 and Supplementary Figs. 13, 15-17, 22, 30. We have also added "Mean values \pm s.d. of triplicate experiments are shown for peak percentages" to their figure legends.

4. FRET traces in the SI rarely show the intermediate FRET state (eg. in Fig S15e). Is this state only found in a subset of traces, or is it stochastically visited?

Response:

We thank the reviewer for the comment. We replaced the smFRET traces in Supplementary Fig. 13i, 15c and 15e to support that the intermediate FRET state can be observed in a subset of traces.

5. The authors attribute different conditions to the different observed state distributions between NMR and FRET experiments. Could the authors estimate, based on K_d , what percentage of state occupancy is expected and match this with the observation?

Response:

We thank the reviewer for the comment. The K_d of riboSAM binding with SAM can be calculated using the equation: $K_d = C_{apo} \times C_{SAM} / (C_{transit} + C_{holo})$, where C_{apo} , C_{SAM} , $C_{transit}$ and C_{holo} are the concentrations of *apo*, SAM, *transit* and *holo* respectively. However, only the knowledge of overall K_d and missing individual K_d for SAM-*transit* and SAM-*holo* binding hinder us from calculating the percentages of individual state. In addition, the concentrations of the RNA immobilized on the slides in smFRET experiments are uncertain. But we did measure K_d of $\sim 28 \mu\text{M}$ from NMR data (Supplementary Table 12), which is comparable with K_d of $\sim 10 \mu\text{M}$ measured in fluorescence study.

6. I got confused about the assignment of the sites in supplementary figure 3: Step 12 supposedly produces EC-81, the first molecule evaluated by smFRET (Fig. S3b). In the scheme S3a, the blue arrow labeled S12 however points at position 75. Position 81 is marked by S13, etc. The authors should double check their labeling in either S3a or S3b, one if incorrect. (S12/S13 arrows in this figure is the start points of the steps, which need to be corrected to avoid confusion)

Response:

We thank the reviewer for pointing it out. In Supplementary Fig. 3, the restart site of step 12 (marked by blue arrow) is position 75, and position 81 is the pause site of step 12, and it is also the restart site of step 13. To avoid the confusion, we added the marks at the pause sites in Supplementary Fig. 3a, and revised its figure legend by adding “The pause sites of steps 12, 13 and 14 are marked by blue dots” in the figure legend.

7. The English requires some further careful editing, in particular in the beginning of the paper.

Response:

We thank the reviewer for the comment. We re-wrote the abstract, some sentences especially in Introduction and NMR section and corrected some grammar errors (please refer to the highlighted at pages 2-13).

REVIEWERS' COMMENTS

Reviewer #1 (Remarks to the Author):

In my opinion the authors have undertaken a great effort to improve the manuscript significantly. All of my original concerns and comments have been addressed. A plethora of new data were accordingly generated. Especially the use of a bacterial polymerase in the additional experiments was strongly requested by all reviewers and is now included in the manuscript. Furthermore, the NMR investigations were extended and are now more sound. I would recommend publication of the manuscript.

Reviewer #3 (Remarks to the Author):

In this revised manuscript, Xue et al made a huge effort to address all open questions. They have added new data to corroborate their findings and they have clarified the text in some parts. I think this is an impressive study, showing the behavior of a riboswitch during the transcription elongation. Regarding the data, I think that they have only minor comments (see below).

However, the text can be sometimes very hard to understand, sometimes due to convoluted sentences and grammatical errors. I think the paper requires professional English editing – this may be provided by the journal.

Minor comments:

- Title: I am not sure that 'growing' captures the system appropriately. In the end, the authors study riboswitch dynamics captures at different steps of transcription. Maybe something like 'Observation of structural dynamics of the SAM-VI riboswitch at different transcription steps at single-nucleotide and single-molecule resolution'.
- The authors should better describe what is shown in Figure 6d,g,h – what is the molecule exactly? The authors only lightly discuss this experiment in the text and it is hard to understand. I would suggest to add a sketch to panels 6d,g showing the hybridized DNA strand.
- Why are the peaks for h'/t' not observable in Figure 6g, as the same structures should be formed?
- New data on polymerases: The authors state that: "However, the peak proportions of EFRET 0.8 are obviously higher in the T7 system than E. coli for EC-86. " Why is this obvious? Is there an explanation why the two systems behave differently (size of the enzyme, for example)?

RESPONSE TO REVIEWERS' COMMENTS

Reviewer #1 (Remarks to the Author):

In my opinion the authors have undertaken a great effort to improve the manuscript significantly. All of my original concerns and comments have been addressed. A plethora of new data were accordingly generated. Especially the use of a bacterial polymerase in the additional experiments was strongly requested by all reviewers and is now included in the manuscript. Furthermore, the NMR investigations were extended and are now more sound. I would recommend publication of the manuscript.

Response: We are deeply grateful for the time and effort that the reviewer put into improving our manuscript.

Reviewer #3 (Remarks to the Author):

In this revised manuscript, Xue et al made a huge effort to address all open questions. They have added new data to corroborate their findings and they have clarified the text in some parts. I think this is an impressive study, showing the behavior of a riboswitch during the transcription elongation. Regarding the data, I think that they have only minor comments (see below).

Response: We are deeply grateful for the time and effort that the reviewer put into improving our manuscript.

However, the text can be sometimes very hard to understand, sometimes due to convoluted sentences and grammatical errors. I think the paper requires professional English editing – this may be provided by the journal.

Response: We thank the reviewer for the comment. We have rewritten a few long sentences and corrected some grammatical errors (highlighted in pages 3, 4, 5, 6, 8, 9, 10, 13, 14 and 17).

Minor comments:

- Title: I am not sure that 'growing' captures the system appropriately. In the end, the authors study riboswitch dynamics captures at different steps of transcription.

Maybe something like 'Observation of structural dynamics of the SAM-VI riboswitch at different transcription steps at single-nucleotide and single-molecule resolution'.

Response: We thank the reviewer for the comment. We agree that “growing” may be not appropriate, and we have changed the title to “Observation of structural switch in nascent SAM-VI riboswitch during transcription at single-nucleotide and single-molecule resolution”.

- The authors should better describe what is shown in Figure 6d,g,h – what is the molecule exactly? The authors only lightly discuss this experiment in the text and it is hard to understand. I would

suggest to add a sketch to panels 6d,g showing the hybridized DNA strand.

Response: We thank the reviewer for the great comment. We have added the sketches to show the hybridized DNA-RNA in Figs. 6d and 6g, and changed the Figure legend accordingly.

- Why are the peaks for h'/t' not observable in Figure 6g, as the same structures should be formed?

Response: We thank the reviewer for the comment. We have added the possible reason to explain the disappearance of peaks h'/t' as “However, the NMR signals h'/t' from U25 are not visible, which may be because U25 in the hybridized ¹⁵N-25U32G-riboSAM is too dynamic to be detected by NMR” (highlighted in page 15).

- New data on polymerases: The authors state that: “However, the peak proportions of EFRET 0.8 are obviously higher in the T7 system than E. coli for EC-86. “Why is this obvious? Is there an explanation why the two systems behave differently (size of the enzyme, for example)?

Response: We thank the reviewer for the comment. We have changed the sentence “However, the peak proportions of E_{FRET} 0.8 are obviously higher in the T7 system than *E. coli* for EC-86” to “However, the peak proportion of E_{FRET} 0.8 in EC-86 is about 8% and 18% higher in the T7 system than *E. coli* in the absence and presence of SAM respectively”. And we have added the possible reasons to explain why the two systems behave differently as “This may be related to the differences between the two RNAP in size, transcription dynamics, interaction mode with DNA and RNA” (highlighted in page 18).